




# Abyssal plain hills and internal wave turbulence


**by Hans van Haren[1]**


[1]Royal Netherlands Institute for Sea Research (NIOZ) and Utrecht University, P.O. Box 59,
1790  AB Den Burg, the Netherlands.
*e-mail: hans.van.haren@nioz.nl



**Abstract.**
A 400-m long array with 201 high-resolution NIOZ temperature sensors was
deployed above a northeast-equatorial Pacific hilly abyssal plain for 2.5 months. The
sensors sampled at 1 Hz, the lowest was at 7 m above the bottom 'mab'. The aim was
to study internal waves and turbulent overturning away from large-scale ocean
topography. Topography consisted of moderate, a few 100 m elevated hills, providing
a mean bottom slope of one-third of that found at the Mid-Atlantic Ridge (on 2 km
horizontal scales). In contrast with observations over large-scale topography like
guyots, ridges and continental slopes, the present data showed a well-defined near-
homogeneous 'bottom-boundary layer' extending between <7 and 100 mab with a
maximum around 65 mab. The average thickness exceeded tidal current bottom-
frictional heights and internal wave breaking dominated over bottom friction. Near-
bottom fronts varied in time (and thus space). Occasional coupling was observed
between the interior internal waves breaking and the near-bottom overturning, with
varying up- and down- phase propagation. In contrast with currents that were dominated
by the semidiurnal tide, 200-m shear was dominant at (sub-)inertial frequencies. The
shear was so large that it provided a background of marginal stability for the straining
high-frequency internal wave field in the interior. Daily averaged turbulence dissipation
rate estimates were between $10^{-10}$ and $10^{-9}$ m$^2$s$^{-3}$, increasing with depth, while eddy
diffusivities were O($10^{-4}$ m$^2$s$^{-1}$). This most intense 'near-bottom' internal wave-induced
turbulence will affect resuspension of sediments.






## 1 Introduction

The mechanical kinetic energy brought into the ocean via tides, atmospheric disturbances and the Earth's rotation governs the motions in the density stratified ocean interior. On the one hand isopycnals are set into oscillating motions as 'internal waves'. On the other hand these oscillating motions deform nonlinearly and eventually irreversibly lose their energy into turbulent mixing: Breaking internal waves are suggested to be the dominant source of turbulence in the ocean (e.g., Eriksen, 1982; Gregg, 1989; Thorpe, 2018). This turbulence is vital for life in the ocean, as it dominates the diapycnal redistribution of components and suspended materials. It is also important for the resuspension of bottom materials. Large-scale sloping ocean bottoms are important for both the generation (e.g., Bell, 1975; LeBlond and Mysak, 1978; Morozov, 1995) and the breaking of internal waves (e.g., Eriksen, 1982). Not only the topography around ocean basin's edges act as source/sink of internal waves but especially also the topography of ridges, mountain ranges and seamounts distributed over the ocean floor. Above sufficiently steep slopes, exceeding those of the main internal carrier (e.g. tidal) wave containing largest energy, and >1 km (> the internal wavelength) horizontal scale topography, turbulent mixing averages 10,000 times molecular diffusion (e.g., Aucan et al., 2006; van Haren and Gostiaux, 2012). This mixing is considered to be 'efficient' as the back and forth sloshing of the carrier wave ensures a rapid restratification down to within a meter from the sea floor, while mixed waters are transported into the interior along isopycnals. Sloping large-scale topography has received more scientific interest than abyssal plains due to the higher turbulence intensity of internal wave breaking. However, it may be questioned whether the abyssal plain and its overlying waters may be called a 'quiescent zone'.





This is because occasional 'benthic storms' have been reported to disturb the
quiescence, even at great depths >5000 m (Hollister and McCave, 1984). The effects
can be great on sediment reworking and particles remain resuspended long after the
'storm' has passed. Such resuspension has obvious effects on deep-sea benthic biology
and remineralization (e.g., Lochte, 1992). In order to avoid semantic problems, the term
'benthic boundary layer' is reserved here for the sediment-water interface (at the bottom
of the water phase of the ocean), following common practice by sedimentologists and
marine chemists (e.g., Boudreau and Jørgensen, 2001). The term 'bottom boundary
layer' follows the physical oceanographic convention to describe the lower part of the
water phase of the ocean which is almost uniform in density, using the threshold
criterion of the large (100-m) scale buoyancy frequency $N < 3 \times 10^{-4}$ s$^{-1}$. This is the layer
of investigation here together with overlying more density stratified waters in the
interior. The amount of homogeneity is also a subject of study. Historic observations
have demonstrated the variability of the abyssal plain bottom boundary layer in space
and time (e.g., Wimbush, 1970; Armi and Millard, 1976; Armi and D'Asaro, 1980).
Similar to the ocean interior, waters above abyssal plains are considered calm ocean
regions in terms of weak turbulent exchange. However, the (bulk) Reynolds number Re
$= UL/\nu$ as a measure for the transition from laminar ('molecular') to turbulent flow is
not small. With the kinematic viscosity $\nu \approx 1.5 \times 10^{-6}$ m$^2$ s$^{-1}$ to characterize the molecular
water properties and characteristic velocity, $U \approx 0.05$ m s$^{-1}$, and length scale, $L \approx 30$ m,
of the (internal wave) water flow, $Re \approx 10^6$, or highly turbulent (e.g., Tennekes and
Lumley, 1972; Fritts et al., 2016) even for the unbounded open ocean and atmosphere
interiors.
Both convective instability of gravitationally unstable denser over less dense water
and shear-induced Kelvin-Helmholtz instability (KHi) are probable for internal wave





breaking, for a recent model see (Thorpe 2018). Earlier models (e.g., Garrett and Munk,
1972) suggested the latter were dominant, especially considering the construction of
the internal wave field of smallest vertical scales residing at the lowest frequencies (e.g.,
LeBlond and Mysak, 1978). Most kinetic energy is found at these frequencies and thus
a large background shear is generated (e.g., Alford and Gregg, 2001) through which
shorter length-scale waves near the buoyancy frequency propagate, break and overturn.
The result is an open ocean wave field that is highly intermittent producing a very
steppy, non-smooth sheet-and-layer-structured ocean interior stratification (e.g., Lazier,
1973; Fritts et al., 2016). In the near-surface ocean, such internal wave propagation and
deformation (straining) of stratification has been observed to migrate through the
density field in space and time.
The lower bound of inertio-gravity wave (IGW) frequencies is determined by the
local vertical Coriolis parameter, i.e. the inertial frequency, $f = 2\Omega\sin\varphi$ of the Earth
rotational vector $\boldsymbol{\Omega}$ at latitude $\varphi$. This bound becomes significantly modified to lower
sub-inertial frequencies under weak stratification ($\sim N^2$), when $N < 10f$, approximately.
From not-approximated equations, minimum and maximum IGW-frequencies are
calculated as $[\sigma_{min}, \sigma_{max}] = (s -\!/\!+ (s^2 - f^2N^2)^{1/2})^{1/2}$ using $2s = N^2 + f^2 + f_h^2\cos^2\gamma$, in which
$\gamma$ is the angle to the north ($\gamma = 0$ denoting meridional propagation) and the horizontal
component of the Coriolis parameter $f_h = 2\Omega\cos\varphi$ becomes important for internal wave
dynamics (e.g., LeBlond and Mysak, 1978; Gerkema et al., 2008).
In the present paper, detailed moored observations from a Pacific abyssal 'plain'
confirm Lazier (1973)'s steppy sheet-and-layer stratification. The new observations are
used to investigate the interplay between motions in the stratified interior and the effects
on the bottom boundary layer. The small-scale topography may prove not negligible in
comparison with large oceanic ridges, seamounts and continental slopes: Following



Bell (1975), recent studies demonstrate the potential of substantial internal wave
generation by flow over abyssal hills under particular slope and stratification conditions
(e.g., Nikurashin et al., 2014; Hibiya et al., 2017). We are interested in observational
details of the IGW-induced turbulent processes.

**2 Data**
Observations were made from the German R/V Sonne above the abyssal hills in the
northeast-equatorial Pacific Ocean, West of the oriental Pacific Ridge (Fig. 1). The area
is not mountainous but also not flat. It is characterized by numerous hills, extending
several 100 m above the surrounding sea floor. The average bottom slope is 1.2±0.6°,
computed from the lower panel of Fig. 1 using the 1'-resolution version of the Smith
and Sandwell (1997) seafloor topography. This slope is about three times larger than
that of the Hatteras plain (the area of observations by Armi and D'Asaro, 1980) and
about three times smaller than that for a similar size area from the Mid-Atlantic Ridge
(West of the Azores). SeaBird SBE911plus CTD profiles were collected 1 km around
11° 50.630′N, 116° 57.938′ W in 4114±20 m water depth at 20-23 March and 06 June
2015. Between 19 March and 02 June a taut-wire mooring was deployed at the above
coordinates. At this latitude, $f = 0.299 \times 10^{-4}$ s$^{-1}$ ($\approx$ 0.4 cpd, cycles per day) and $f_h =$
$1.427 \times 10^{-4}$ s$^{-1}$ ($\approx$ 2 cpd). A 130 m elevation has its ridge at approximately 5 km West
of the mooring.
The mooring consisted of 2700 N of net top-buoyancy at about 450 m from the
bottom. With current speeds of less than 0.15 m s$^{-1}$, the buoy did not move more than
0.1 m vertically and 1 m horizontally, as was verified using pressure and tilt sensors.
The mooring line held three single point Nortek AquaDopp acoustic current meters, at
6, 207 and 408 mab, meters above the bottom. The middle current meter was clamped





to a 0.0063 m diameter plastic coated steel cable. To this 400 m long insulated cable
201 custom-made 'NIOZ4' temperature sensors were taped at 2.0 m intervals. To
deploy the 400 m long instrumented cable it was spooled from a custom-made large-
diameter drum with separate 'lanes' for T-sensors and the cable (Appendix A).

The NIOZ4 T-sensor noise level is <0.1 mK, the precision <0.5 mK (van Haren et

al., 2009; NIOZ4 is an update of NIOZ3 with similar characteristics). The sensors
sampled at a rate of 1 Hz and were synchronized via induction every 4 h, so that their
timing mismatch was <0.02 s and the 400 m profile was measured nearly
instantaneously. As in the abyssal area temperature variations are extremely small,
severe constraints were put on the de-spiking and noise levels of data. Under these
constraints, 35 (17% of) T-sensors showed electronic timing, calibration or noise
problems. Their data are no longer considered and are linearly interpolated. This low-
biases estimates of turbulence parameters like dissipation rate and diffusivity from T-
sensor data by about 10%. Appendix B describes further data processing details.

During three days around the time of mooring deployment and two days after

recovery, shipborne conductivity-temperature-depth (CTD) profiles were made for
monitoring the temperature-salinity variability from 5 m below the surface to 10 mab.
A calibrated SeaBird 911plus CTD was used. The CTD data were processed using the
standard procedures incorporated in the SBE-software, including corrections for cell
thermal mass using the parameter setting of Mensah et al. (2009) and sensor time-
alignment. All other analyses were performed with Conservative (~potential)
Temperature ($\Theta$), absolute salinity SA and density anomalies $\sigma_4$ referenced to 4000
dbar using the GSW-software described in (IOC, SCOR, IAPSO, 2010).

After establishment of the temperature-density relationship (Appendix B), the

moored T-sensor data are used to estimate turbulence dissipation rate $\varepsilon = c_1^2 d^2 N^3$ and



vertical eddy diffusivity $K_z = m_1 c_1^2 d^2 N$ following the method of reordering potentially
unstable vertical density profiles in statically stable ones, as proposed by Thorpe
(1977). Here, d denotes the displacements between unordered (measured) and reordered
profiles. N denotes the buoyancy frequency computed from the reordered profiles. We
use standard constant values of $c_1 = 0.8$ for the Ozmidov/overturn scale factor and $m_1$
$= 0.2$ for the mixing efficiency (Osborn, 1980; Dillon, 1982; Oakey, 1982). The validity
of the latter is justified after inspection of the temperature-scalar spectral inertial
subrange content (cf. Section 3) and also considering the generally long averaging
periods over many (>1000) profiles. The buoyancy Reynolds number $Re_b = \varepsilon/\nu N^2$ is
used to distinguish between areas of weak, $Re_b < 100$, and strong turbulence.
In the following, averaging over time is denoted by […], averaging over depth-
range by <…>. The specific averaging periods and ranges are indicated with the mean
values. The vertical coordinate z is taken upward from the bottom z = 0. Shear-induced
overturns are visually identified as inclined S-shapes in log(N) panels while convection
demonstrates more vertical columns (e.g., van Haren and Gostiaux, 2012; Fritts et al.,
2016). It is noted that both types occur simultaneously, as columns exhibit secondary
shear along the edges and KHi demonstrate convection in their interior core (Li and Li,
2004; Matsumoto and Hoshino, 2006).

## 203    3 Observations

High-resolution T-sensor data analysis was difficult because of the very small
temperature ranges and variations of only a few mK over, especially the lower, 100 m
of the observed range. This rate of variation is less than the local adiabatic lapse rate.
First, a spectral analysis is performed to investigate the internal wave and turbulence
ranges and slopes appearance. Then, particular turbulent overturning aspects of internal





wave breaking are demonstrated in magnifications of time-depth series. Finally,
profiles of mean turbulence parameter estimates are used to focus on the extent and
nature of the bottom boundary layer.

.

**3.1 Spectral overview**

The small temperature ranges are reflected in the low values of the large-scale

stratification (Fig. 2a). (Salinity contributes weakly to density variations, Appendix B).
Typical buoyancy periods are 3 h, increasing to roughly 9 h in near-homogeneous
layers, e.g., near the bottom. In spite of the weak stratification, the IGW-band,
approximately between and including f and N, is one order of magnitude wide. This
IGW-bandwidth is observable in spectra of turbulence dissipation rate (Fig. 2b) and
temperature variance (Fig. 2c).

The T-sensors have identical instrumental noise levels at frequencies $\sigma > 10^4$ cpd

and near-equal variance at sub-inertial frequencies $\sigma < f$ (Fig. 2c). In the frequency
range in between, and especially for $f \prec \sigma \prec N$, the upper T-sensor data demonstrate
largest variance by up to two orders of magnitude at $\sigma \approx N$ compared with the lower T-
sensor data. In this frequency range, the upper T-sensor spectrum has a slope of about
-1 (in the log-log domain), which reflects a dominance of smooth quasi-linear ocean-
interior IGW (van Haren and Gostiaux, 2009). Extending above this slope is a small
near-inertial peak reflecting rarely observed low internal wave frequency vertical
motions in weakly stratified waters (van Haren and Millot, 2005). The steep -3 roll-off
at super-buoyancy frequencies $\sigma > N$ is also associated with IGW. At frequencies in
between, and for the lower T-sensor data throughout the frequency range, a slope of -
5/3 is found. This reflects passive scalar turbulence dominated by shear (Tennekes and
Lumley, 1972). After sufficient averaging this passive scalar turbulence is efficient





(Mater et al., 2015). At intermediate depth levels, and in short frequency ranges of the
spectral data, slopes vary between -2 and -1. Slopes between -5/3 and -1 would point at
active scalar turbulence of convective mixing (Cimatoribus and van Haren, 2015) while
a slope of -2 reflects finestructure contamination (Phillips, 1971) or a saturated IGW-
field (Garrett and Munk, 1972).

While the upper T-sensor data contain most variance and hence most potential

energy in the IGW-band, the spectrum of estimated turbulence dissipation rate
demonstrates nearly two orders of magnitude higher variance for the lowest T-sensor
data around $\sigma \approx f$ (Fig. 2b). The stratification around the upper sensor supports
substantial internal waves, but weak turbulence provides a flat and featureless spectrum
of the dissipation rate time series. The lower layer $\varepsilon$-spectrum shows a relative peak
near $\sigma \approx 2f$, but no peaks at the inertial and semidiurnal tidal frequencies. The lack of
peaks at the latter frequencies is somewhat unexpected as the kinetic energy (Fig. 2b,
blue spectrum) is highly dominated by motions at $M_2$ and, to a lesser extent, at just
super-inertial 1.04f.

In contrast, the 'large-scale shear' spectrum computed between current meters 20 m

apart (Fig. 2b, light-blue) shows a single dominant peak at just sub-inertial 0.99f, with
a complete absence of a tidal peak. This reflects large quasi-barotropic vertical length
scales >400 m exceeding the mooring range at semidiurnal tidal frequencies and
commonly known 'small' ≤200 m vertical length scales at near-inertial frequencies.
The large-scale shear has an average magnitude of $<|\mathbf{S}|> = 2 \times 10^{-4}$ s$^{-1}$ for 207-408 mab
and $1.6 \times 10^{-4}$ s$^{-1}$ for 6-207 mab, with peak values of $|\mathbf{S}| = 6 \times 10^{-4}$ s$^{-1}$ and $4 \times 10^{-4}$ s$^{-1}$,
respectively. Considering mean $<N> \approx 5.5 \times 10^{-4}$ s$^{-1}$ with variations over one order of
magnitude, the mean gradient Richardson number $Ri = N^2/|\mathbf{S}|^2$ is just larger than unity
while marginally stable conditions ($Ri \approx 0.5$; Abarbanel et al., 1984) occur regularly.



Unfortunately, higher vertical resolution (acoustic profiler) current measurements were
not available to establish smaller scale shear variations associated with higher
frequency internal waves propagating through the (large-scale) shear generated by the
near-inertial motions. Such smaller-scale variations in shear are expected in association
with sheet-and-layer variation in stratification observed using the detailed high-
resolution T-sensors.

**3.2 Detailed periods**

The days shortly after deployment were amongst the quietest in terms of turbulence

during the entire mooring period. Nevertheless, some near-bottom and interior turbulent
overturning was observed occasionally (Fig. 3). For this example, averages of
turbulence parameters for one day time interval and 400 m vertical interval are
estimated as $[<\varepsilon>] = 1.2\pm0.8\times10^{-10}$ m$^2$ s$^{-3}$ and $[<K_z>] = 7\pm4\times10^{-5}$ m$^2$ s$^{-1}$. These values
are typical for open-ocean 'weak turbulence' conditions although mean $Re_b \approx 200$.
Shortest isotherm distances are observed far (a few 100 m) above the bottom (Fig. 3a)
reflecting the generally stronger stratification (Fig. 3b) there. While the upper isotherms
smoothly oscillate with a periodicity close to the average buoyancy period of 3.2 h and
amplitudes of about 15 m, the stratification is organized in fine-scale layering
throughout, except for the lower 50 m of the range. Detailed inspection of sheets (large
values of small-scale $N_s$ in Fig. 3b) demonstrates that they gain and lose strength 'strain'
over time scales of the buoyancy period and shorter, that they merge and deviate, e.g.
around 300 mab between days 82.25 and 82.5 in Fig. 3b, also from the isotherms, in
association with the largest turbulent overturns (Fig. 3c) eroding them. This is reflected
in non-smooth isotherms, e.g., the interior overturning near 220 mab and day 82.6. The
patches of interior turbulent overturns, with displacements $|d| < 10$ m in this example,



are elongated in time-depth space, having timescales of up to the local buoyancy period
but not longer. Thus, it is unlikely they represent an intrusion that can have timescales
exceeding the local buoyancy timescale. Considering the 0.05 m s$^{-1}$ average (tidal)
advection speed, their horizontal spatial extent is estimated to be about 500 m. This
extent is very close to the estimated baroclinic Rossby radius of deformation $Ro_i$ =
NH/nπf ≈ 600 m for vertical length scale H = 100 m and first mode n = 1.

The near-bottom range is different, with buoyancy periods approaching the

semidiurnal period, sometimes longer. However, a permanent turbulent and
homogeneous 'bottom boundary layer' is not observed, after further detailing (Fig. 4).

Examples of the upper, middle and lower 100 m of the T-sensor range are presented

in magnifications with different colour range, while maintaining the same isotherm
interval of 5 mK (Fig 4a-c). For this period, the mean flow is 0.04±0.01 m s$^{-1}$ towards
the SE, more or less off-slope of the small ridge located 5 km West of the mooring.
Between these panels, the high-frequency internal wave variations decrease in
frequency from upper to lower, but all panels do show overturning (e.g., around 330
mab and day 82.35 in Fig. 4a, 200 mab and day 82.6 in Fig. 4b and 35 mab and day
82.5 in Fig. 4c). In Fig. 4c the entire T-colour range represents only 1 mK. In this depth-
range, the low-frequency variation in temperature and, while not related, stratification
vary with a period of about 15.5 h. These variations do not have tidal periodicity and
are thus not reflecting bottom friction of the dominant tidal currents. Quasi-convective
overturning seems to occur after day 82.5. In the interior > 100 mab most overturning
seems shear-induced.

The overturning phenomena are more intensely observed during a less quiescent

day (Figs 5, 6), when turbulence values are about five times larger and mean $Re_b$ ≈
1400. Between 300 and 400 mab isotherms remain quite smooth with near-linear





internal wave oscillations (Fig. 5a,b). The lower 300 m are quasi-permanently in
turbulent overturning but in specific bands only around about 310 mab and around 160
mab (Fig. 5c,d). Rms vertical overturn displacements are 2-3 times larger than in the
previous example. Their duration is commensurate the local buoyancy periods. The
smooth upper range isotherms centered around 360 mab are reflected in 75 m, one day
wide range of turbulence dissipation rates below threshold (Fig. 5d). But, above and
especially below, turbulent overturning is more intense, see also the detailed panels
(Fig. 6). While shear-induced overturning is seen, e.g. around 200 mab day 97.95 (Fig.
6b), convective turbulence columns are observed e.g. around 60 mab and day 98 (Fig.
6c).0. It is noted however, that in the presented data we cannot distinguish the fine-
detailed secondary overturning, e.g. shear-induced billow formation, on convection
'vertical columns'. In the lower 100 mab, overturning occurs on the large (~50 m,
hours) scales but also on much shorter time scales of 10 min. This results in isotherm
excursions that are faster than further away from the bottom. A coupling between
interior and near-bottom (turbulence and internal wave) motions is difficult to establish.
For example, short-scale (high-frequency $\sigma \gg f$) internal wave propagation >200 mab
shows downward phase (i.e., upward energy) propagation around day 97.75 (Fig. 5a)
with no clear correspondence with the lower 100 mab. Between days 98.2 and 98.5
however, the phase propagation appears upward (downward energy propagation), with
some indication for correspondence between upper 200-400 mab and lower 100 mab.
During this period the mean 0.04 m s$^{-1}$ flow was towards W (upslope).

Another example of (two days) of rather intense turbulence is given in Figs 7,8,

with similar average values as in the previous example. It demonstrates in particular
relatively large-amplitude near-N internal waves (e.g., day 112.9, 310 mab) and bursts
of elongated weakly sloping (slanting) shear-induced overturning (e.g., day 113.2, 210



mab). The near-N waves appear quasi-solitary lasting for maximum 2 periods and
having about 30 m trough-crest level variation. As before, the vertical phase
propagation of these waves is ambiguous. In addition, very high-frequency 'internal
waves' around the small-scale buoyancy frequency are observed in the present
example, with small amplitudes <10 m visible in the isotherms around 300 mab, day

113.1.

The interior turbulent overturning appears more intense than in preceding examples,
with larger excursions of about |50 m| near 200 mab (Fig. 8b). This slanting layer of
elongated overturns seems originally shear-induced, but the overturns show clear
convective properties during the observed stage. The largest duration of patches is close
to the local mean buoyancy period. The entire layer demonstrates numerous shorter
time-scale overturning. Cross-overs (sudden changes in the vertical) are observed of
isotherms from thin high-$N_s$ above low-$N_s$ turbulent patches to below the low-$N_s$
patches, e.g. day 112.6 in Fig. 7b, and vice-versa, e.g. days 113.1 and 113.5 (recall that
small-scale $N_s$ is computed from reordered $\Theta$-profiles). This evidences one-sided,
rather than two-sided, turbulent mixing eroding a stratified layer either from below or
above.
The interior shear-induced turbulent overturning seems to have some
correspondence with the (top of) the near-bottom layer: on days 113.1-113.6 interior
mixing is accompanied by similar near-bottom mixing. The status of the near-bottom
layer ($z < 75$ mab) switches from large-scale convective instabilities (day < 113.1) to
stratified shear-induced overturning ($113.1 <$ day $< 113.6$) and back to large-scale
convection with probably secondary shear instabilities (day > 113.6). This is visible in
the displacements (Fig. 7c) and dissipation rate (Fig. 7d), and part of it in detailed
temperature (Fig. 8c). The transitions between near-bottom 'mixing regimes' are



abruptly marked by near-bottom fronts. The mean 0.03 m s$^{-1}$ flow is SW-directed (more
or less on-slope).
A two-day example of a relatively intensely turbulent near-bottom layer is given in
Figs 9,10. Two periods of about 9 h long (around days 135.9 and 136.8), 22 h apart,
demonstrate >50 m tall convective overturning extending nearly 100 mab. In between,
large-scale shear-induced overturning dominates, with a possible correspondence with
the interior in the form of a large-scale doming of isotherms and mixing in patches
around day 135.4 (lasting between 135.25 < day < 135.75, generally around 200 mab).
The doming interior isotherms are not repeated in the lowermost isotherm capping of
the near-bottom layer, except perhaps for the down-going flank/front. The mean NE-
flow is 0.03 m s$^{-1}$ (more or less off-slope). In this example as well as in previous ones
no evidence is found for 'smooth' intrusions, as demonstrated in the atmospheric DNS-
model by Fritts et al. (2016).

**3.3 Mean profiles**
The different mixing observed in the interior and near the bottom is reflected in the
'mean profiles' of estimated turbulence parameters (Fig. 11a-c). These plots are
constructed from patching together consecutive one-day portions of data that are locally
drift-corrected. Time-average values of [ε], turbulent flux (providing average [$K_z$]) and
stratification (providing average [N]) are computed for each depth level. Averaging
over a day and longer is exceeding the buoyancy period even in these weakly stratified
waters. It is thus considered appropriate for internal wave induced mixing. This may
lead to some counter-intuitive averaging of displacement values greater than the local
distance to the bottom at particular depths. However, it is noted that Prandtl's concept
of overturn sizes never exceeding the distance to a solid boundary was based on

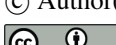



turbulent friction of flow over a flat plate. As Tennekes and Lumley (1972) indicate,
such 'mixing length theoretical concept' may not be valid for flows with more than one
characteristic velocity. The present area is not known for geothermal fluxes, which are
also not observed in the present data. Here, the dominant turbulence generation process
seems induced by internal waves, as the observed turbulence well extends above the
layer O(10 m) of bottom friction.

The mean dissipation rate (Fig. 11a) and diffusivity (Fig. 11b) profiles are observed

to be largest between 7 and 60 mab, with values at least ten times higher than in the
interior. Near the bottom, stratification (Fig. 11c) is low but not as weak as some 15 m
higher-up. At about 30 mab local minima of $[\varepsilon]$ and $[K_z]$ are found. The average top of
weakly stratified $N < 3 \times 10^{-4}$ s$^{-1}$ $\approx$ 4 cpd 'bottom boundary layer' is at about 65 mab
(Fig. 11d). This sub-maximum in the pdf-distribution is broader than a second
maximum closer to the bottom, near 10 mab. This smaller bottom boundary layer is
probably induced by current friction, whereas the larger layer with an average of 65
mab probably by internal wave turbulence. Around 110 mab the maximum of the
bottom boundary layer is found with few occurrences (Fig. 11d). Around that height,
the profiles' minimum turbulence values are observed at the depth of a weak local
maximum N (Fig. 11c). This layer separates the interior turbulent mixing with
maximum around 200 mab and the 'near-bottom' (<100 mab) mixing. From the
detailed data in Section 3.2 correspondence is observed between these layers, occurring
at least occasionally. Considering the weaker (mean) turbulence in between, it is
expected that the correspondence is communicated via internal waves and their shear.
As for freely propagating IGW, its frequency band has a one order of magnitude width
nearly everywhere, also close to the bottom (Fig. 11c). It is noted that inertial waves
from all (horizontal) angles can propagate through homogeneous, weakly and strongly



stratified layers, thus providing local shear (LeBlond and Mysak, 1978; van Haren and
Millot, 2004).

**4 Discussion**
The observed turbulence at 100 m and higher above the sea floor is mainly induced
by (sub-)inertial shear and (small-scale) internal wave breaking. This confirms
suggestions by Garrett and Munk (1972) for interior IGW. However, this shear is not
found to be decreasing with N (depth) in the present data. The >100 mab depth range
is termed 'the interior' here although perhaps not being representative for the 'mid-
water ocean' as it is still within the height range of surrounding hilly topography. The
130 m high ridge 5 km West of the mooring is well outside the baroclinic Rossby radius
of deformation ($Ro_i \approx 500$ m). It unlikely influences the near-bottom turbulence here,
also because no correlation is found between across-slope flow and turbulence
intensity. The interior is occasionally found quiescent, with parameter values below the
threshold of very weak turbulence at about ten times molecular diffusion values. More
commonly the interior is found weakly-moderately turbulent with values
commensurate with open-ocean values (e.g., Gregg, 1989) following the interaction of
high-frequency internal waves breaking and inertial shear.
The observed dominance of near-inertial shear at the 200 m vertical scale, the
vertical separation distance between the current meters, is found far below the depths
of atmospheric disturbances generation near the surface. It seems related with local
generation, possibly in association with the hilly topography (St. Laurent et al., 2012;
Nikurashin et al., 2014; Alford et al., 2016; Hibiya et al., 2017). Also, the 200 m vertical
scale is observed to well exceed the excursion length (amplitude) of the internal waves,
the scale of overturn displacements and the size of most density stratification layering.



In contrast, above the Mid-Atlantic Ridge, where tidal currents are only twice as
energetic as near-inertial motions, the vertical length scale of tides equals that of near-
inertial motions around about 100-150 m (van Haren, 2007). There and in the open
ocean, near-inertial motions dominate shear at shorter scales with an expected peak
around 25 m (e.g., Gregg, 1989). As in the present data, the near-inertial shear showed
a shift to sub-inertial frequencies (van Haren, 2007). As the shear-magnitude was found
to be concentrated in sheets of high-N, it was suggested that this red-shift was due to
the broadening of the IGW-band in low-N layers. As a result, an effective coupling
between shear, stratification and the IGW-band was established. Considering the
similarity in sheet-and-layering and (large-scale) shear, such coupling is also suggested
in the present observations from the deep-sea over less dramatic topography.

As for a potential coupling between the interior and the more intense near-bottom

turbulence, internal wave propagation is observed in both up and down directions. In
the lower 50 mab the variability in turbulence intensity, in turbulence processes of shear
and convection, and in stratification demonstrates a non-smooth bottom boundary layer,
an active near-bottom turbulent zone 'NBTZ'. As observed by Armi and D'Asaro
(1980), the extent above the bottom of turbulent mixing and a near-homogeneous mixed
layer varies between <7 and 100 mab with a mean of about 65 mab. This mean value
exceeds the common frictional boundary scales that can be computed for flow over flat
bottoms on a rotating sphere (Ekman, 1905), although parametrizations provide one
order of magnitude differences: $\delta = (2A/f)^{1/2}$, A the turbulent viscosity; if taken $A = K_z$
$\approx 10^{-4}\text{-}10^{-3}$ m$^2$ s$^{-1}$; Fig. 11b, $\delta \approx 2.5\text{-}8$ m, or $\delta = 2 \times 10^{-3}U/f$; $U \approx 0.05$ m s$^{-1}$: $\delta \approx 30$ m
(e.g., Tennekes and Lumley, 1972). Both are (substantially) less than the NBTZ found
here, which thus seems to be governed by other processes such as IGW-breaking.





Sloping fronts are observed near the bottom in Armi and D'Asaro (1980)'s, Thorpe
(1983)'s and the present data. However, isopycnal transport of mixed waters seems not
away from the boundary as proposed in (Armi and D'Asaro, 1980) but rather into the
NBTZ sloping downward with time (present data). This governs the variable height of
the NBTZ.
Although bottom slopes were about three times larger in the Northeast Pacific than
above the Hatteras Plain, the present observations show many similarities as in Armi
and D'Asaro (1980). They also show many similarities with equivalent turbulence
estimates in both the interior and in the variable lower 100 mab compared with those
from above the central Alboran Sea, a basin of the Mediterranean Sea (van Haren,
2015), and with observations made in the southeast Pacific abyssal hill plains around -
07° 07.213′ S, -088° 24.202′ W, East of the oriental Pacific Ridge (unpublished results).
Thus it seems that the precise characteristics (slopes/heights) of the hilly topography is
not very relevant for the observed internal wave intensity and turbulence generation, as
long as the bottom is not a flat plate and the hills have IGW-scales. This probably holds
for both the present observations in the stratified interior and those in the NBTZ. The
tenfold larger turbulence intensity in the latter marks a relatively extended inertial
subrange. Although the near-bottom (6 mab) current magnitudes are typically 0.05 m
s$^{-1}$, up to about 0.10 m s$^{-1}$, the estimated turbulence intensity of $10^3$-$10^4$ times larger
than molecular diffusion is sufficient to mix materials up to 100 mab, the extent of
observed vertical mixing in the layer adjacent ot the bottom. This reflects previous
observations of nephels, turbid waters of enhanced suspended materials (Armi and
D'Asaro, 1980). It is expected that this material is resuspended locally, as the more
intensely turbulent steeper large-scale slopes are too far away horizontally, far beyond
the baroclinic Rossby radius of deformation.




For the future, modelling may provide better insights in the precise coupling
between near-inertial shear and internal wave breaking, leading to a combination of
convective and shear-induced overturning. The one-sided shear across thin-layer
stratification, as inferred from observed deviation of high-N sheets from isotherms and
associated with the vertical propagation direction of internal waves, may prove
important for the wave breaking.

**490    5 Conclusions**

From the present high-resolution temperature sensor data moored up to 400 m above
a hilly abyssal plain in the northeastern Pacific we find an interaction between small-
scale internal wave propagation, large-scale near-inertial shear and the near-bottom
water phase. In an environment where semidiurnal tidal currents dominate, 200-m shear
is largest at the inertial frequency and near-bottom turbulence dissipation rates are
largest at twice the inertial frequency. Due to internal wave propagation and occasional
breaking, stratification in the overlying waters is organized in thin sheets, with less
stratified waters in larger layers in between, but turbulent erosion occurs
asymmetrically. The average amount of turbulent overturns due to internal wave
breaking here and there is equal to open ocean turbulence, with intensities about 100
times those of molecular diffusion. The high-frequency internal waves propagate to
near the bottom and likely trigger  ten times larger turbulence there as shown in time-
average vertical profiles. The result is a highly variable near-bottom turbulent zone,
which may be near-homogeneous over heights of less than 7 m and up to 100 m above
the bottom. This near-bottom turbulence is not predominantly governed by frictional
flows on a rotating sphere as in Ekman dynamics that occupy a shorter range O(10 m)
above the bottom. Fronts occur and sudden isotherm-uplifts by solitary internal waves



508 as well. Turbulence seems shear dominated, but occurs in parallel with convection. The

509 shear is quasi-permanent because the dominant near-circular inertial motions have a

510 constant magnitude. It is expected that inertial shear dominates also on shorter scales

511 (not verifiable with the present current meter data), possibly added by smaller internal

512 wave shear. In the mean, turbulence dissipation rate exceeds the level of $10^{-11}$ $m^2s^{-3}$,

513 except for a 30 m thick layer around 100 mab.

515 *Acknowledgements.* I thank the master and crew of the R/V Sonne, J. Greinert and A.

516 Vink for their pleasant contributions to the overboard sea-operations. J. Blom

517 meticulously welded the thermistor string drums, including all of the pins. Financial

518 support came from the Netherlands Organization for the Advancement of Science

519 (N.W.O.), under grant number ALW-856.14.001 (JPIOceans).





APPENDIX A

**Thermistor string drum: A dedicated instrumented cable spool**

The deployment of a 1D T-sensors mooring, a thermistor string, is like most commonly done for oceanographic moorings. Through the aft A-frame the top-buoy is put first in the water whilst the ship is slowly steaming forward. The thermistor string is coupled between buoy/other instrument(s) and other instrument(s)/acoustic releases before attaching the weight that is dropped in free fall. The thermistor string is put overboard through a wide, relatively large (0.4 m) diameter pulley, about 2 m above deck, or, preferably, via a smoothly rounded gunwhale (Fig. A1). Up to 100 m length of string holding typically 100 T-sensors can be put overboard manually by one or two people. In that case, the string is laid on deck in neat long loops. The deployment of a longer length string becomes more difficult, because of the weight and drag. For such strings a 1.48 m inner diameter (1.60 m OD) 1400 pins drum is constructed to safely and fully control their overboard operation (Fig. A1). The drum dimensions fit in a sea container for easy transportation. The 0.04 m high metal pins guide the cables and separate them from the T-sensors in 'lanes', while allowing the cables to switch between lanes. The pins are screwed and welded in rows 0.027 and 0.023 m apart, the former sufficiently wide to hold the sensors. Up to 18 T-sensors can be located in one lane, before the next lane is filled. The drum has 14 double lanes and can store about 230 T-sensors and 450 m of cable in one layer. The longest string deployed successfully thus far held 300 T-sensors and was 600 m long, with about one-quarter of the string doubled on the drum. The doubling did not pose a problem, the sensors were thus well separated that entanglement did not occur. For recovery, or deployment of strings holding up to 150 T-sensors, a smooth surface drum is used of the same dimensions but without pins.



APPENDIX B

**Temperature sensor data processing in weakly stratified waters**
High-resolution T-sensors can be used to estimate vertical turbulent exchange
across density-stratified waters, under particular constraints that are more difficult to
account for under weakly stratified conditions of $N < 0.1f$, say. As in the present data
the full temperature range is only 0.05℃ over 400 m, careful calibration is needed to
resolve temperatures well below the 1-mK level, at least in relative precision.
Correction for instrumental electronic drift of 1-2 mK/mo requires shipborne high-
precision CTD knowledge of the local conditions and uses the physical condition of
static stability of the ocean at time scales longer than the buoyancy scale (longer than
the largest turbulent overturning timescale). CTD knowledge is also needed to use
temperature data as a tracer for density variations.
The NIOZ4 T-sensor noise level is nominally $<1\times10^{-4}°C$ (van Haren et al., 2009;
NIOZ4 is an update of NIOZ3 with similar characteristics) and thus potentially of
sufficient precision. A custom-made laboratory tank can hold up to 200 T-sensors for
calibration against an SBE35 Deep Ocean Standards high precision platinum
thermometer to an accuracy of $2\times10^{-4}°C$ over ranges of about 25°C in the domain of [-
4, +35]°C. Due to drift in the NTC-resistor and other electronics of the T-sensors, such
accuracy can be maintained for a period of about four weeks after aging. However, this
period is generally shorter than the mooring period (of up to 1.5 years). During post-
processing, sensor-drifts are corrected by subtracting constant deviations from a smooth
profile over the entire vertical range and averaged over typical periods of 4-7 days.
Such averaging periods need to be at least longer than the buoyancy period to guarantee
that the water column is stably stratified by definition (in the absence of geothermal
heating as in the present area). Conservatively, they are generally taken longer than the





inertial period (here: 2.5 days). In weakly stratified waters as the present observations,
the effect of drift is relatively so large that the smooth polynomial is additionally forced
to the smoothed CTD-profile obtained during the overlapping time-period of data
collection (Fig. B1). In the present case, this can only be done during the first few days
of deployment and corrections for drift during other periods are made by adapting the
local smooth polynomial with the difference of the (smooth-average) CTD-profile and
the smooth polynomial of the first few days of deployment.
The calibrated and drift-corrected T-sensor data are transferred to Conservative
(~potential) Temperature ($\Theta$) values (IOC, SCOR, IAPSO, 2010), before they are used
as a tracer for potential density variations $\delta\sigma_4$, referenced to 4000 dbar, following the
constant linear relationship obtained from best-fit data using all nearby CTD-profiles
over the mooring period and across the lower 400 m (Fig. B2). As temperature
dominates density variations, this relationship's slope or apparent thermal expansion
coefficient is $\alpha = \delta\sigma_4/\delta\Theta = -0.223\pm0.005$ kg m$^{-3}$ °C$^{-1}$ (n=5). The resolvable turbulence
dissipation rate threshold averaged over a 100-m vertical range is approximately $3\times10^{-12}$ m$^2$ s$^{-3}$.




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





**Figure 1**. Bathymetry map of the tropical Northeast Pacific based on the 9.1 ETOPO-
1 version of satellite altimetry-derived data by Smith and Sandwell (1997). The
black dot in the lower panel indicates mooring and CTD positions. Note the different
colour ranges between the panels.

**Figure 2**. Stratification and spectral overview. (**a**) Vertical profiles of buoyancy
frequency scaled with the local horizontal component of the Coriolis parameter $f_h$
and smoothed over 50 dbar (~50 m), from all five CTD-stations to within 1 km from
the mooring. The blue, green and red profiles are made around the time of mooring
deployment. (**b**) Weakly smoothed (10 degrees of freedom, dof) spectra of kinetic
energy (upper current meter; green) and current difference (between upper and
middle current meters; light-blue). In red and purple the spectra of 150 s sub-
sampled time series of 100 m vertically averaged turbulence dissipation rates for
lower (7-107 m above the bottom, mab) and upper (307-407 mab) T-sensor data
segments, respectively. The inertial frequency f, $f_h$ including several higher
harmonics, buoyancy frequency N incl. range, and the semidiurnal lunar tidal
frequency $M_2$ are indicated. $N_{max}$ indicates the maximum small-scale buoyancy
frequency. (**c**) Weakly smoothed (10 dof) spectra of 2 s sub-sampled temperature
data from 3 depths representing upper, middle and lower levels. For reference,
several slopes with frequency are indicated.

**Figure 3**. One day sample detail of moored temperature observations during relatively
calm conditions (on the day of calibration in the beginning of the record). (**a**)
Conservative Temperature. The black contour lines are drawn every 0.005°C. At
the top from left to right, two time references indicate the mean (purple bar) and





shortest (green bar) buoyancy periods found in this data-detail. Values for time-

depth-range-mean parameters are given of buoyancy Revnolds number (light-blue),

buoyancy frequency (blue), turbulence dissipation rate (red) and turbulent eddy

diffusivity (black). Errors for the latter two are to within a factor of 2,

approximately. (**b**) Logarithm of small-scale (2 dbar) buoyancy frequency from

reordered temperature profiles. The black isotherms are reproduced from panel a.

(**c**) Thorpe displacements between raw-(panel a.) and reordered T-profiles. (d)

Logarithm of turbulence dissipation rate.

**Figure 4**. Magnifications of Fig. 3a using different colour ranges but maintaining the 5

mK distance between isotherms. (**a**) Upper 100 m. (**b**) Approximately middle 100

m. (**c**) Bottom 100 m; note the entire colour range extending over 1 mK only. (d)

Time series of logarithm of vertical-mean turbulence dissipation rates from Fig. 3d

for the panels a,b,c labelled u,m,b, respectively.

**Figure 5**. As Fig. 3 with identical colour ranges, but for a one-day period with more

intense turbulence especially near the bottom.

**Figure 6**. As Fig. 4, but associated with Fig. 5 and using different colour ranges.

**Figure 7**. As Fig. 3 with identical colour ranges, but for a two-day period with

occasional long shear turbulence.

**Figure 8**. As Fig. 4, but associated with Fig. 7 and using different colour ranges.





**Figure 9**. As Fig. 3 with identical colour ranges, but for a two-day period with some
intense convective turbulence also near the bottom.

**Figure 10**. As Fig. 4, but associated with Fig. 9 and using different colour ranges.

**Figure 11**. Profiles of turbulence parameters from entire-record time-averaged
estimates using 1-day drift-corrected, 150 s sub-sampled moored temperature data.
(**a**) Logarithm of dissipation rate. (**b**) Logarithm of eddy diffusivity. (**c**) Logarithm
of small-scale (2 dbar) buoyancy frequency from the T-sensors (black) with for
comparison the mean of the five CTD-profiles smoothed over 50 dbar vertical
intervals from Fig. 2a (red). The green dashed curves indicate the minimum (to the
left of the f-line) and maximum (to the right of the N-profile) inertio-gravity wave
bounds for meridional internal wave propagation (see text). (d) Pdf of the 'bottom
boundary layer height', the level of the first passage of threshold $N > 3 \times 10^{-4}$ $s^{-1}$
indicating the stratification capping the 'near-homogeneous' layer from the bottom
upward. Two peaks are visible, one near 10 mab attributable to bottom friction,
another around 65 mab attributable to internal wave-induced turbulence.

**Fig. A1**. Photo of thermistor string deployment using the instrumented cable spooling
drum onboard R/V Sonne.

**Fig. B1**. Conservative Temperature profiles with depth over the lower 400 mab. One-
day mean moored sensor data, raw data after calibration (thin black line, yellow-
filled) and smooth high-order polynomial fit (thick black solid line). In red are three
CTD-profiles within 1 km from the mooring during the first days of deployment



(two solid profiles on day 80/81 coincide in time with moored data mean), in blue-
dashed are two CTD-profiles after recovery of the mooring. The mean of the two
solid red profiles is given by the red/dash-dot profile, 0.015 ℃ off-set for clarity,
with its smooth high-order polynomial fit in light-blue to which the moored data are
corrected.

**Fig. B2**. Lower 400 m of five CTD-profiles obtained near the T-sensor mooring. Red
data are from around the beginning of the moored period, blue from after recovery.
(**a**) Conservative Temperature. (**b**) Absolute Salinity with x-axis range matching the
one in a. in terms of equivalent relative contributions to density variations. The noise
level is larger than for temperature. (**c**) Density anomaly referenced to 4000 dbar.
(**d**) Density anomaly – Conservative Temperature relationship ($\delta\sigma_4 = \alpha\delta\Theta$). The
data yielding two representative slopes after linear fit are indicated (the mean of 5
profiles gives $<\alpha> = -0.223\pm0.005$ kg m$^{-3}$ °C$^{-1}$).




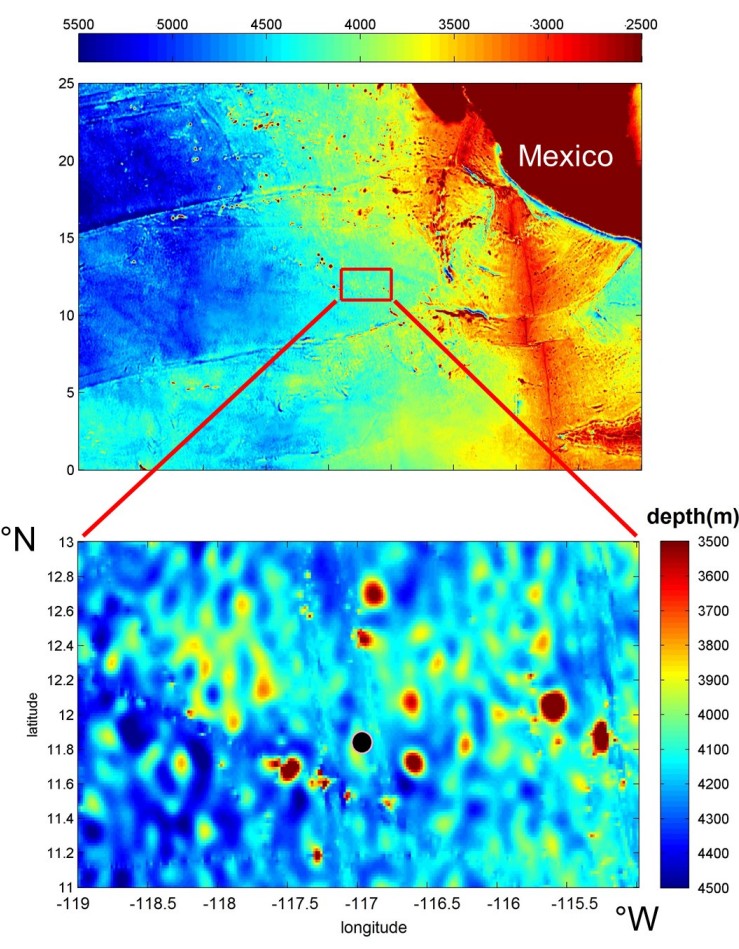

**Figure 1**. Bathymetry map of the tropical Northeast Pacific based on the 9.1
ETOPO-1 version of satellite altimetry-derived data by Smith and Sandwell (1997).
The black dot in the lower panel indicates mooring and CTD positions. Note the
different colour ranges between the panels.



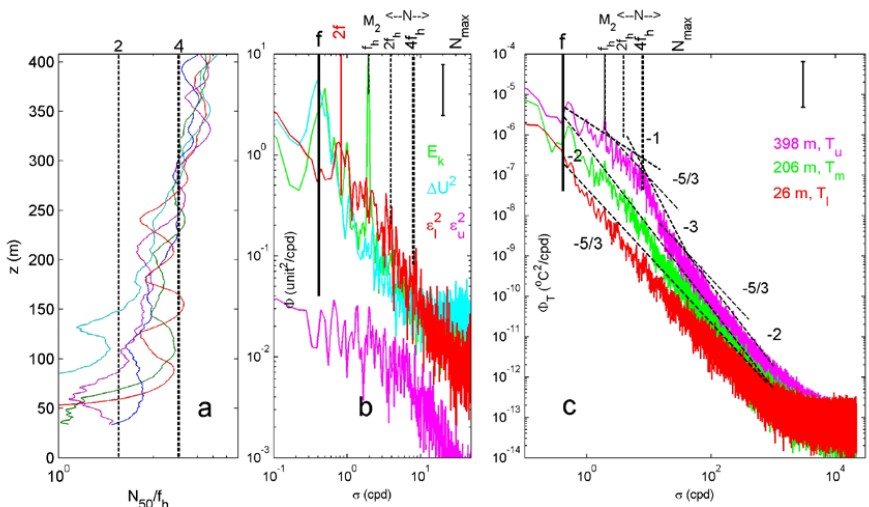

**Figure 2**. Stratification and spectral overview. (a) Vertical profiles of buoyancy
frequency scaled with the local horizontal component of the Coriolis parameter $f_h$
and smoothed over 50 dbar (~50 m), from all five CTD-stations to within 1 km from
the mooring. The blue, green and red profiles are made around the time of mooring
deployment. (b) Weakly smoothed (10 degrees of freedom, dof) spectra of kinetic
energy (upper current meter; green) and current difference (between upper and
middle current meters; light-blue). In red and purple the spectra of 150 s sub-
sampled time series of 100 m vertically averaged turbulence dissipation rates for
lower (7-107 m above the bottom, mab) and upper (307-407 mab) T-sensor data
segments, respectively. The inertial frequency $f$, $f_h$ including several higher
harmonics, buoyancy frequency $N$ incl. range, and the semidiurnal lunar tidal
frequency $M_2$ are indicated. $N_{max}$ indicates the maximum small-scale buoyancy
frequency. (c) Weakly smoothed (10 dof) spectra of 2 s sub-sampled temperature
data from 3 depths representing upper, middle and lower levels. For reference,
several slopes with frequency are indicated.





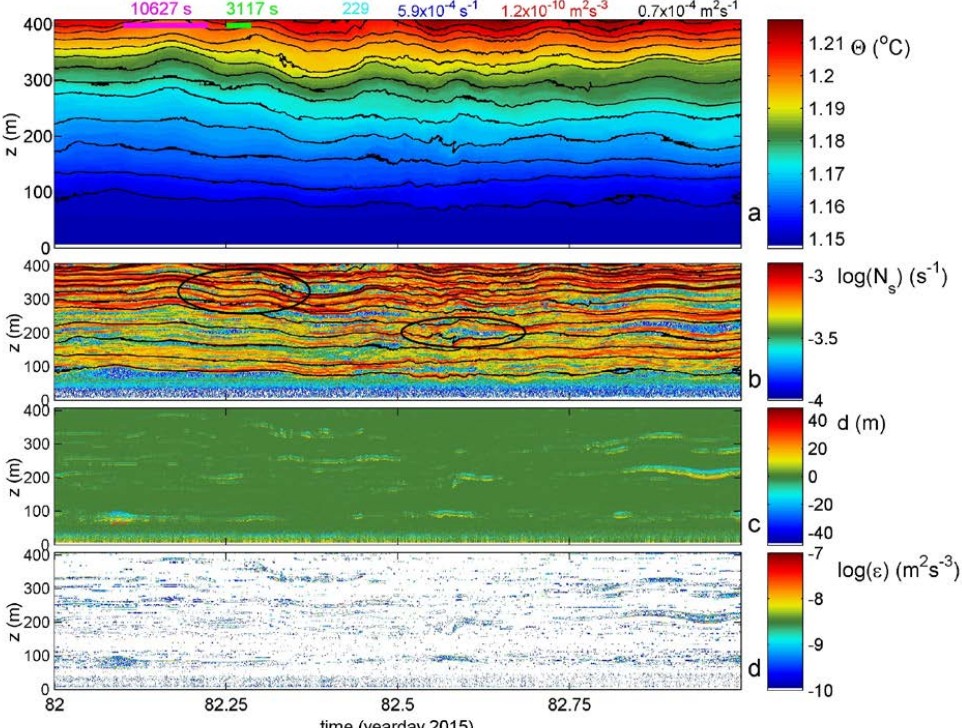

**Figure 3**. One day sample detail of moored temperature observations during relatively calm conditions (on the day of calibration in the beginning of the record). (a) Conservative Temperature. The black contour lines are drawn every 0.005°C. At the top from left to right, two time references indicate the mean (purple bar) and shortest (green bar) buoyancy periods found in this data-detail. Values for time-depth-range-mean parameters are given of buoyancy Revnolds number (light-blue), buoyancy frequency (blue), turbulence dissipation rate (red) and turbulent eddy diffusivity (black). Errors for the latter two are to within a factor of 2, approximately. (b) Logarithm of small-scale (2 dbar) buoyancy frequency from reordered temperature profiles. The black isotherms are reproduced from panel a. (c) Thorpe displacements between raw-(panel a.) and reordered T-profiles. (d) Logarithm of turbulence dissipation rate.



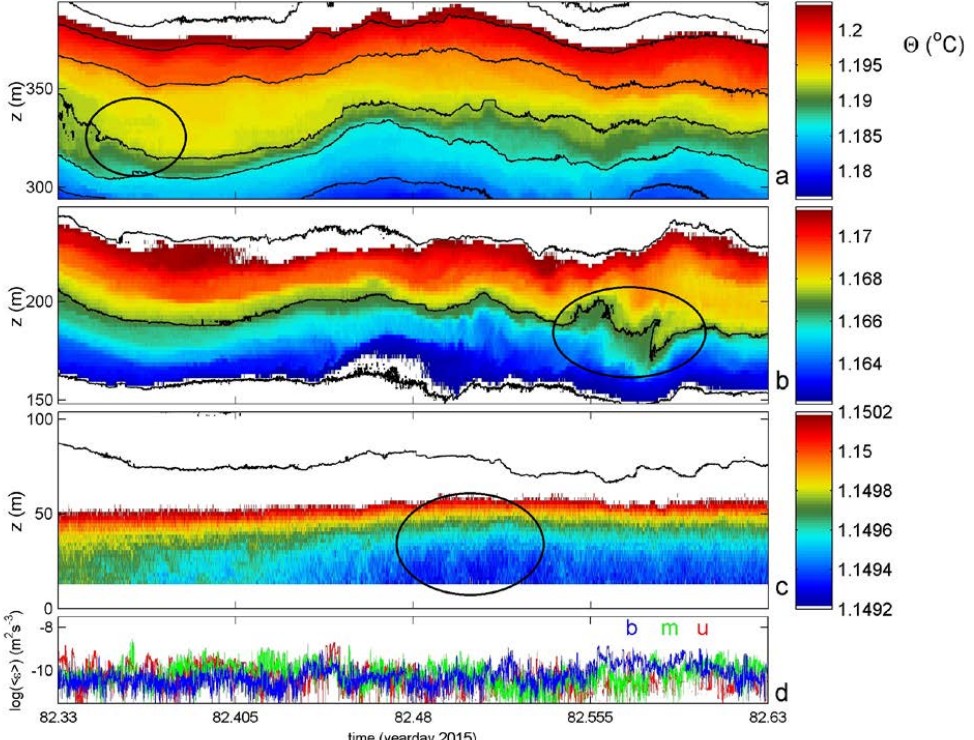

822
**Figure 4**. Magnifications of Fig. 3a using different colour ranges but maintaining the 5 mK distance between isotherms. (a) Upper 100 m. (b) Approximately middle 100 m. (c) Bottom 100 m; note the entire colour range extending over 1 mK only. (d) Time series of logarithm of vertical-mean turbulence dissipation rates from Fig. 3d for the panels a,b,c labelled u,m,b, respectively.



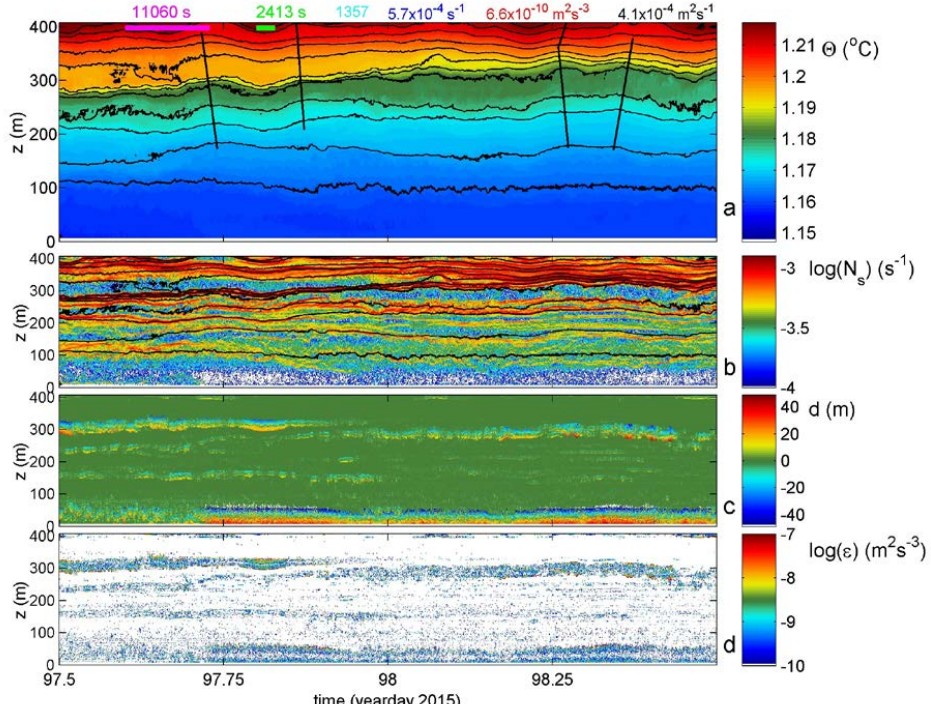

**Figure 5**. As Fig. 3 with identical colour ranges, but for a one-day period with
more intense turbulence especially near the bottom.





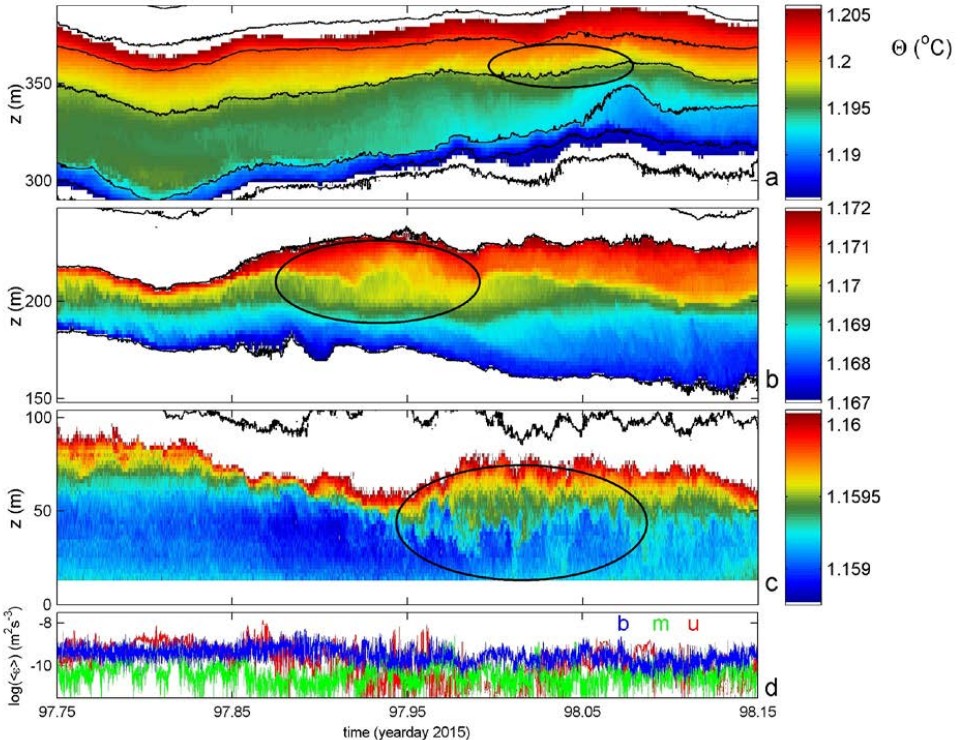

**Figure 6**. As Fig. 4, but associated with Fig. 5 and using different colour ranges.



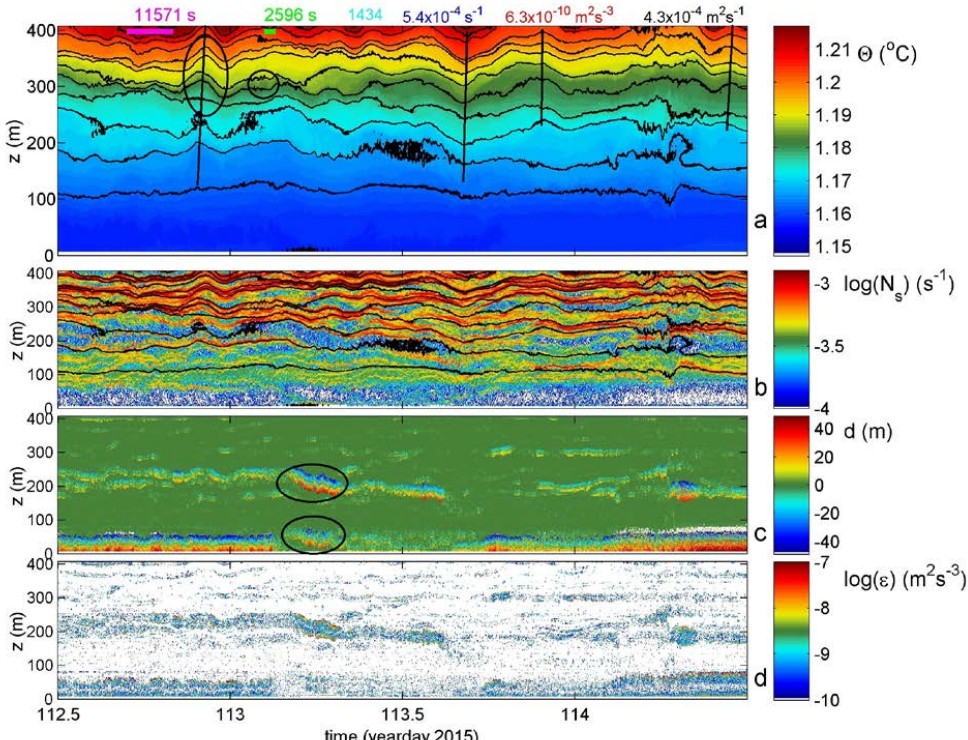

**Figure 7**. As Fig. 3 with identical colour ranges, but for a two-day period with
occasional long shear turbulence.





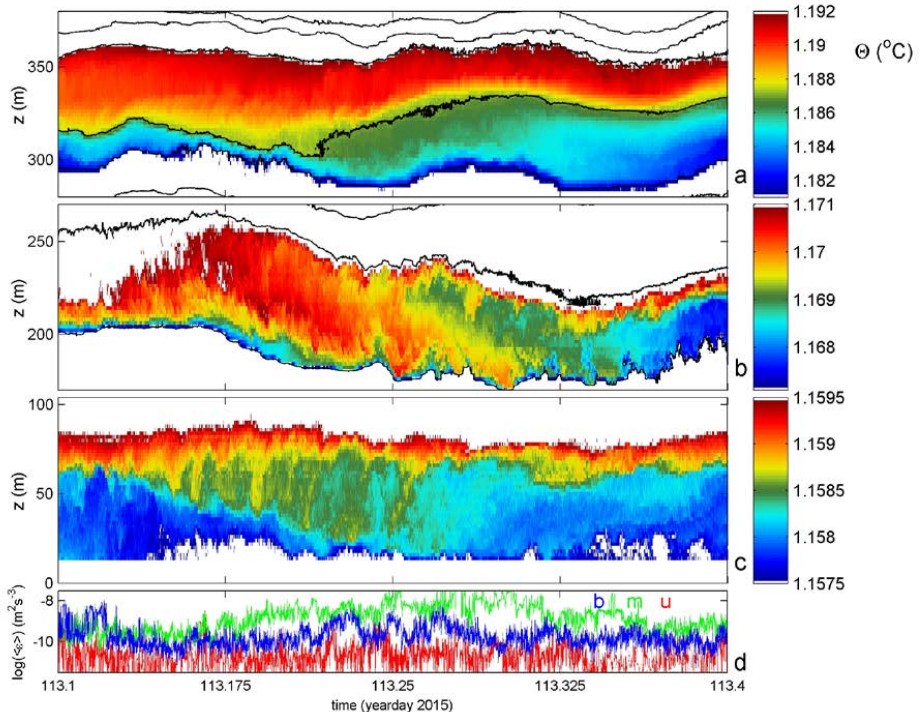

**Figure 8**. As Fig. 4, but associated with Fig. 7 and using different colour ranges.



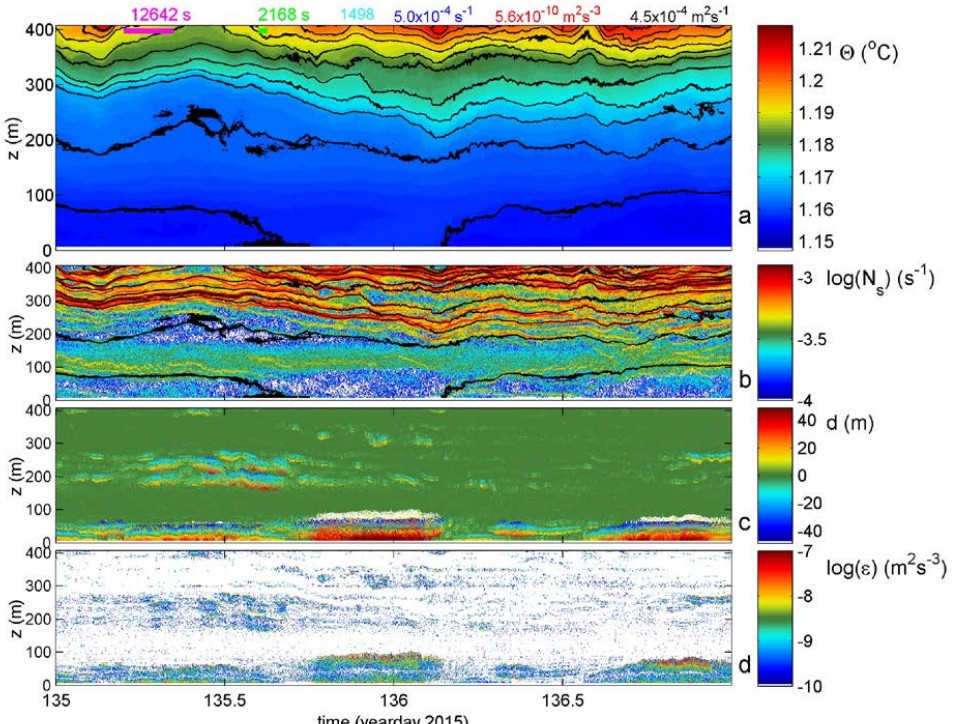


**Figure 9**. As Fig. 3 with identical colour ranges, but for a two-day period with
some intense convective turbulence also near the bottom.



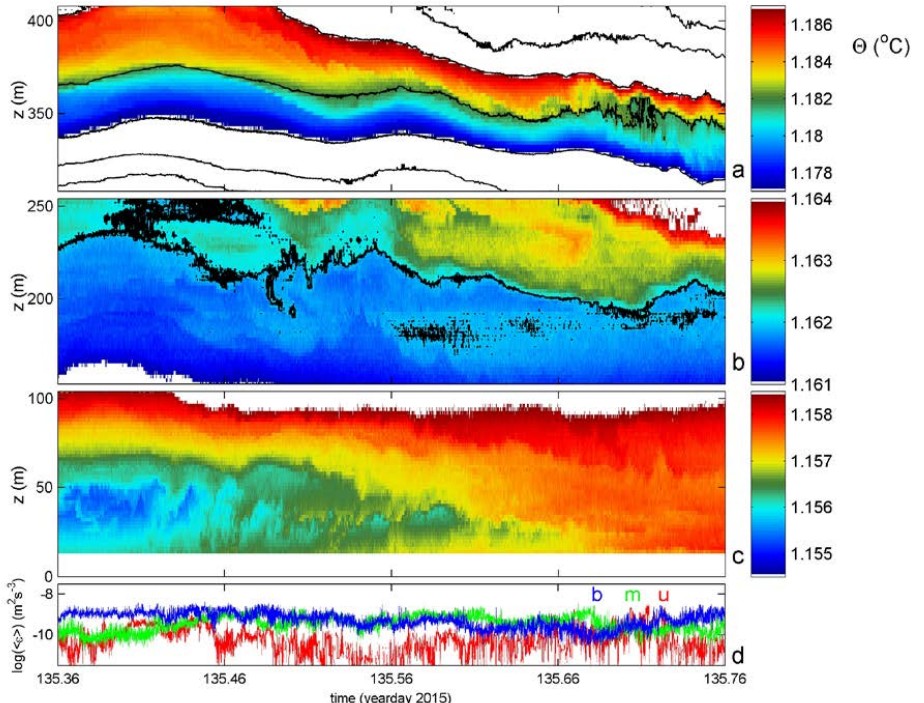

**Figure 10**. As Fig. 4, but associated with Fig. 9 and using different colour ranges.





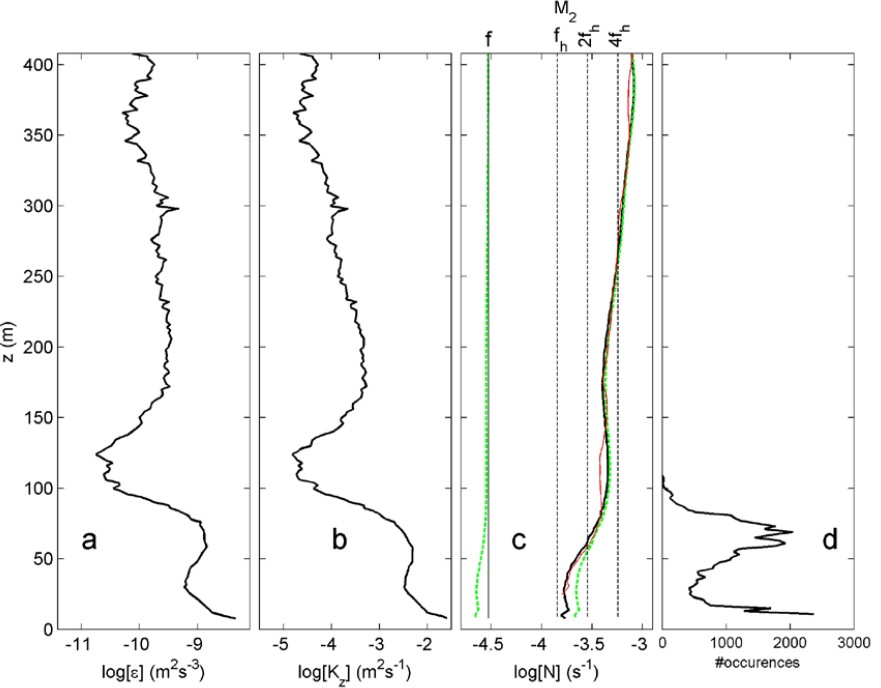

**Figure 11**. Profiles of turbulence parameters from entire-record time-averaged estimates using 1-day drift-corrected, 150 s sub-sampled moored temperature data. (a) Logarithm of dissipation rate. (b) Logarithm of eddy diffusivity. (c) Logarithm of small-scale (2 dbar) buoyancy frequency from the T-sensors (black) with for comparison the mean of the five CTD-profiles smoothed over 50 dbar vertical intervals from Fig. 2a (red). The green dashed curves indicate the minimum (to the left of the f-line) and maximum (to the right of the N-profile) inertio-gravity wave bounds for meridional internal wave propagation (see text). (d) Pdf of the 'bottom boundary layer height', the level of the first passage of threshold $N > 3 \times 10^{-4}$ s$^{-1}$ indicating the stratification capping the 'near-homogeneous' layer from the bottom upward. Two peaks are visible, one near 10 mab attributable to bottom friction, another around 65 mab attributable to internal wave-induced turbulence.




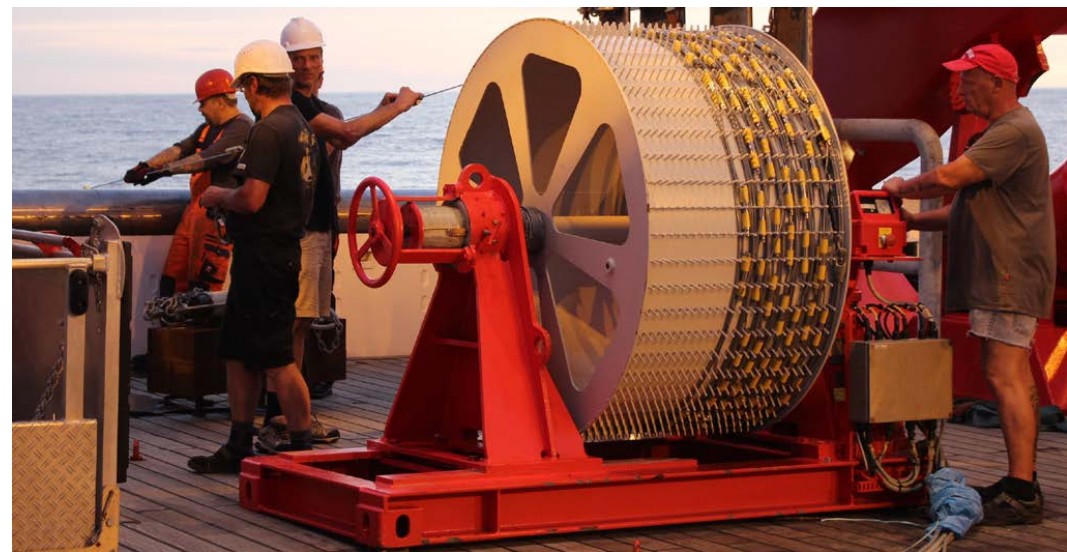

**Fig. A1**. Photo of thermistor string deployment using the instrumented cable spooling drum onboard R/V Sonne.





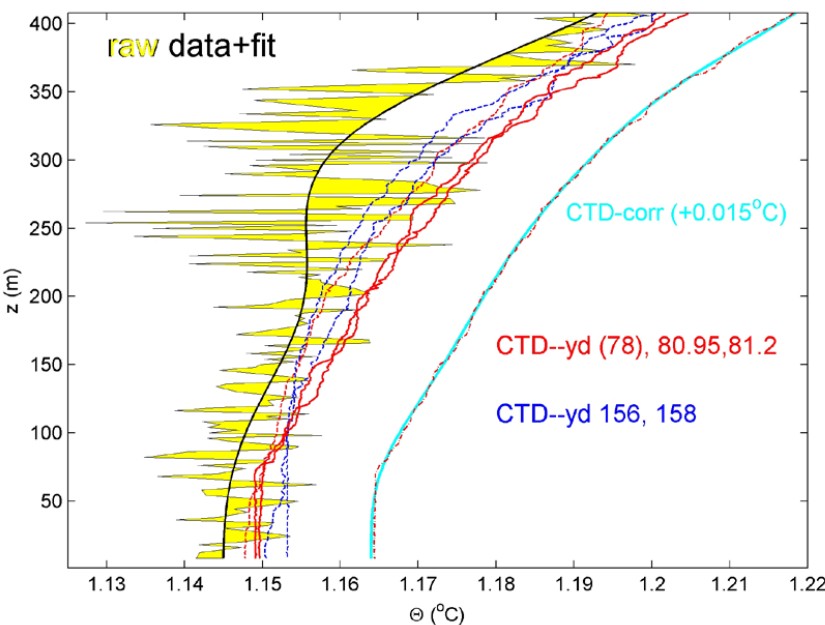

**Fig. B1**. Conservative Temperature profiles with depth over the lower 400 mab. One-day mean moored sensor data, raw data after calibration (thin black line, yellow-filled) and smooth high-order polynomial fit (thick black solid line). In red are three CTD-profiles within 1 km from the mooring during the first days of deployment (two solid profiles on day 80/81 coincide in time with moored data mean), in blue-dashed are two CTD-profiles after recovery of the mooring. The mean of the two solid red profiles is given by the red/dash-dot profile, 0.015 ℃ off-set for clarity, with its smooth high-order polynomial fit in light-blue to which the moored data are corrected.





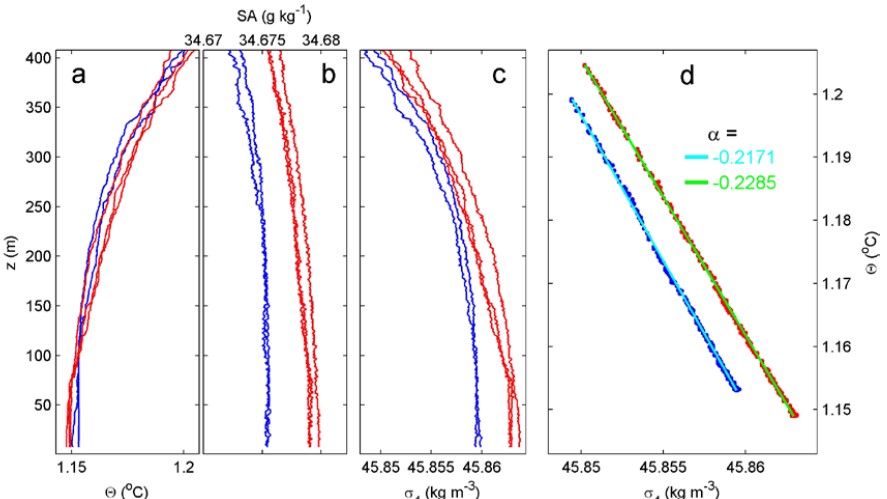

**Fig. B2**. Lower 400 m of five CTD-profiles obtained near the T-sensor mooring. Red data are from around the beginning of the moored period, blue from after recovery. (a) Conservative Temperature. (b) Absolute Salinity with x-axis range matching the one in a. in terms of equivalent relative contributions to density variations. The noise level is larger than for temperature. (c) Density anomaly referenced to 4000 dbar. (d) Density anomaly – Conservative Temperature relationship ($\delta\sigma_4 = \alpha\delta\Theta$). The data yielding two representative slopes after linear fit are indicated (the mean of 5 profiles gives $\langle\alpha\rangle$ = -0.223±0.005 kg m$^{-3}$ °C$^{-1}$).