# Peer review of "Abyssal plain hills and internal wave turbulence"

_Biogeosciences, 2018_

## Referee Comment (RC1) · E. Morozov (Referee) · 16 Apr 2018

E. Morozov (Referee)

egmorozov@mail.ru

Review of the manuscript by H. van Haren Abyssal plain hills and internal wave turbulence

The author uses a line of high-resolution temperature sensors to find the interaction between small scale internal waves and large-scale shear near the bottom. Owing to the existence of internal waves and their breaking the stratification exists in thin stratified sheets and thicker layers between them. A highly variable near-bottom turbulent zone was found. Occasional solitary waves uplift the isotherms.

I know from the publications by van Haren that the NIOZ temperature sensors (many of them in a vertical line) are an important tool to study small scale processes in the ocean (line 169).

[Figure]

Line 70 The author should cite his own important publications publication on the measurements in the Romanche FZ and Kane FZ: H. van Haren et al., Convective and shear-induced turbulence in the deep Kane Gap, J. Geophys. Res: V. 118, p. 5924–5930; H. van Haren et al., Extremely long Kelvin-Helmholtz billow trains in the Romanche Fracture Zone, Geophys. Res. Lett., Vol. 41, 2014, p. 8445-8451.

line 74 Please cite one of the most comprehensive publications on internal wave generation by seamounts distributed over the ocean floor.[Baines PG (2007) Internal tide generation by seamounts. Deep Sea Res 54(9):1486–1508]

line 80 citation: Sloping large-scale topography has received more scientific interest than abyssal plains due to the higher turbulence intensity of internal wave breaking.

However, abyssal plains occupy a large part of the ocean and the processes that occur there deserve investigation. A contribution to these studies was made in Morozov 2018 in the regions of the hills in the Gambia Abyssal Plain, Madagascar Basin, and deep Pacific. In some of the regions the small hills on the bottom form corrugated topography instead of the seemingly flat bottom and contribute to internal wave generation and breaking.

line 132 I absolutely agree that: The small-scale topography may prove not negligible in comparison with large oceanic ridges, seamounts and continental slopes. This statement should be pronounced throughout the entire text.

line 141 I suggest that the authors indicates longitudes in the upper panel of Fig. 1. In addition. A region north of the one analyzed in the manuscript was studied in [Morozov 2018].

line 694 I believe ETOPO and Smith&Sandwell are similar but different databases. They require different citations.

line 814 Figure 3 Reynolds not Revnolds number

line 823 Figure 4. Please clarify that upper 100 are the upper 100 m of your line of

thermistors not upper 100 m of the ocean.

Please explicitly indicate what processes are highlighted with black ellipses in the figures. Just number them and explain the process they highlight.

I recommend the manuscript for publication after minor revision.

Please also note the supplement to this comment:
https://www.biogeosciences-discuss.net/bg-2018-142/bg-2018-142-RC1-supplement.pdf

---

## Author Comment (AC1) · 19 Apr 2018

»>I thank the reviewer for the time taken and for the comments made. My replies are behind »>

Review of the manuscript by H. van Haren Abyssal plain hills and internal wave turbulence

The author uses a line of high-resolution temperature sensors to find the interaction between small scale internal waves and large-scale shear near the bottom. Owing to the existence of internal waves and their breaking the stratification exists in thin stratified sheets and thicker layers between them. A highly variable near-bottom turbulent zone was found. Occasional solitary waves uplift the isotherms.

I know from the publications by van Haren that the NIOZ temperature sensors (many

of them in a vertical line) are an important tool to study small scale processes in the ocean (line 169).

»>Thank you for the appreciation

Line 70 The author should cite his own important publications publication on the measurements in the Romanche FZ and Kane FZ: H. van Haren et al., Convective and shear-induced turbulence in the deep Kane Gap, J. Geophys. Res: V. 118, p. 5924–5930; H. van Haren et al., Extremely long Kelvin-Helmholtz billow trains in the Romanche Fracture Zone, Geophys. Res. Lett., Vol. 41, 2014, p. 8445-8451.

»>Those observations represent internal wave and shear-driven turbulence some distance from the bottom in through-flows. The results form a contrast with the present observations, because the near-bottom zone is stratified there despite relatively strong shear flow. It is similar with internal wave convection penetrating to close to the bottom, occasionally. Will mention this now, but later in the discussion.

line 74 Please cite one of the most comprehensive publications on internal wave generation by seamounts distributed over the ocean floor.[Baines PG (2007) Internal tide generation by seamounts. Deep Sea Res 54(9):1486–1508]

»>Yes of course, with the notion that Baines' modeling suggests that seamounts are of the some order (although less than) important for internal tide generation compared with continental slopes/Hawaiian Ridge. It also noted that Baines considered mounts with vertical scale height h > 1 km, but the result may be transferrable to the perhaps even larger amount of hills, with h < 1 km.

line 80 citation: Sloping large-scale topography has received more scientific interest than abyssal plains due to the higher turbulence intensity of internal wave breaking.

However, abyssal plains occupy a large part of the ocean and the processes that occur there deserve investigation. A contribution to these studies was made in Morozov 2018 in the regions of the hills in the Gambia Abyssal Plain, Madagascar Basin, and deep

[Figure]

Pacific. In some of the regions the small hills on the bottom form corrugated topography instead of the seemingly flat bottom and contribute to internal wave generation and breaking.

»>Yes I agree, cited now.

line 132 I absolutely agree that: The small-scale topography may prove not negligible in comparison with large oceanic ridges, seamounts and continental slopes. This statement should be pronounced throughout the entire text.

»>Mentioned now in several places

line 141 I suggest that the authors indicates longitudes in the upper panel of Fig. 1.

»>OK done

In addition. A region north of the one analyzed in the manuscript was studied in [Morozov 2018].

»>OK

line 694 I believe ETOPO and Smith&Sandwell are similar but different databases. They require different citations.

»>Yes, correct, modified now, as the S&S version Topo_9.1b was used.

line 814 Figure 3 Reynolds not Revnolds number »>thank you

line 823 Figure 4. Please clarify that upper 100 are the upper 100 m of your line of thermistors not upper 100 m of the ocean.

»>Done now

Please explicitly indicate what processes are highlighted with black ellipses in the figures.

»>Thank you for pointing out, the locations were given in the text, but the coupling to

the ellipses should have been given directly, as is done now.

I recommend the manuscript for publication after minor revision.

---

## Referee Comment (RC2) · Anonymous Referee #2 · 22 May 2018

Review to Hans van Haren Abyssal plain hills and internal wave turbulence Biogeosciences Discussions

Thank you for reporting on this interesting data set. The paper deals with an observed high-resolution temperature timeseries above an abyssal hill region. It describes internal wave, stratification, overturn, and estimated mixing aspects, as derived by frequency spectra, Thorpe scales, and exemplary showcases. Important points of outcome are a diagnosed relatively intense mixing, particularly in the bottom boundary layer (BBL), as well as a proposed mechanism causing this (internal waves propagating from above into a marginal stable bottom boundary layer and triggering instability). The data set is unique and from an interesting setting between steep and smooth topography, and away from mainstream focus. It merits to be publicly visible, although in the present form I would not recommend to publish the paper. The two main reasons for that are reproducibility, and a possible flaw in the Thorpe scale analysis that would

depreciate major results of the paper.

Reproducibility: a range of results which are specified in the abstract and conclusions sections are not based on data analysis by objective methods or are not treated in the results or discussion sections (a methods part is entirely missing). E.g.: - The coupling mechanism/interaction/interplay between internal waves above the bottom boundary layer and their effects within the BBL. - Sediment resuspension. - Internal wave breaking to be the dominant cause for forming the BBL. - Evidence for the occurrence of fronts and solitary internal waves. - Asymmetric turbulent erosion of stratified layers. - Abundances of overturns. - Turbulence to be caused by both shear instability and convection alike.

Thorpe scale analysis: There is a striking pattern in the calculated Thorpe displacements, indicating a very frequent and long-lasting 50m overturn at the lowest 50m. I assume this is an artefact, because the temperature gradient is often at or below 0.5mK/50m, and sensor noise and uncertainty are comparatively high. In such a constellation of a very low density gradient like in the BBL, noise/uncertainty will cause spurious overturns and overestimated displacements, leading to overestimated mixing through Thorpe scale analysis [Piera et al., 2002; Johnson and Garrett, 2004]. The diagnosed intense mixing in the bottom 50m is at the base of major results of the paper: the increasing turbulence with depth, intense near-bottom mixing, and the explanation for the intense near-bottom mixing by internal waves which trigger overturns in a marginal stable regime. Given the importance of the intense bottom boundary layer mixing for the paper, a critical review of the appropriateness of the used Thorpe scale processing should be a central part of the methods. If the existence of a quasi-permanently overturning 50m-bottom-layer should prove true, this as well should be a central part of the discussions.

Further remarks: - Given the reported numbers N = 5.5 * 10 ˆ -4, S = 1.6 * 10 ˆ -4 (lines 254 to 258), the average Richardson number in the BBL seems rather 10 than unity. This would not support the assumption of the BBL being systematically marginally

stable. - Can you make clearly understandable why the given arguments (lines 189 to 192) allow to choose a mixing efficiency parameter m of 0.2? - I'd propose to more prominently place the particular results for the abyssal hill region in the larger context of the limiting cases 'steep topography' and 'smooth abyssal plain' - data availability is not stated

References:

Piera, Roget, Catalan (2002): Turbulent patch identification in microstructure profiles: a method based on wavelet denoising and Thorpe displacement analysis, J. Atm. Oceanic Tech., 19, 1390-1402

Johnson and Garrett (2004): Effects of noise on Thorpe scales and run lengths, J. Phys. Oceanogr., 34, 2359-2372

---

## Author Comment (AC2) · 5 Jun 2018

»>I thank the reviewer for the time taken to comment my ms. Replies are behind »>

The author uses a line of high-resolution temperature sensors to find the interaction between small scale internal waves and large-scale shear near the bottom. Owing to the existence of internal waves and their breaking the stratification exists in thin stratified sheets and thicker layers between them. A highly variable near-bottom turbulent zone was found. Occasional solitary waves uplift the isotherms.

I know from the publications by van Haren that the NIOZ temperature sensors (many of them in a vertical line) are an important tool to study small scale processes in the ocean (line 169). »>Thank you for the appreciation, indeed the near-bottom zone is highly variable in turbulence.

[Figure]

Thank you for reporting on this interesting data set. The paper deals with an observed high-resolution temperature timeseries above an abyssal hill region. It describes internal wave, stratification, overturn, and estimated mixing aspects, as derived by frequency spectra, Thorpe scales, and exemplary showcases. Important points of outcome are a diagnosed relatively intense mixing, particularly in the bottom boundary layer (BBL), as well as a proposed mechanism causing this (internal waves propagating from above into a marginal stable bottom boundary layer and triggering instability).

The data set is unique and from an interesting setting between steep and smooth topography, and away from mainstream focus. It merits to be publicly visible, although in the present form I would not recommend to publish the paper. The two main reasons for that are reproducibility, and a possible flaw in the Thorpe scale analysis that would depreciate major results of the paper. »>Indeed, the abyssal hills areas are not generally studied. What does the reviewer mean by 'reproducibility'? As outlined in the previous version, and below, there is no flaw in the analysis as the reviewer suggests. It is stressed that the moored chain of high-resolution T-sensors is not the same as 'standard' shipborne CTD profiling.

Reproducibility: a range of results which are specified in the abstract and conclusions sections are not based on data analysis by objective methods or are not treated in the results or discussion sections (a methods part is entirely missing). E.g.: - The coupling mechanism/interaction/interplay between internal waves above the bottom boundary layer and their effects within the BBL. - Sediment resuspension. - Internal wave breaking to be the dominant cause for forming the BBL. - Evidence for the occurrence of fronts and solitary internal waves. - Asymmetric turbulent erosion of stratified layers. - Abundances of overturns. - Turbulence to be caused by both shear instability and convection alike. »>I am puzzled what is meant here. A methods part is not at all missing! Yes, a section named 'Methods' did not exist in the previous version, but data handling was (and is) described: Section 2 Data, and Appendices A and B gave/give instrumental and methods details. Section 2 is now elaborated somewhat (re. the comment below) and it is now called 'Methods and data handling'. I refute that specifications in abstract and conclusions are not treated in the results and discussion sections. I do not recognize the general summing up given by the reviewer. It would have been helpful if clear examples from the manuscript were given (e.g., by indicating line-numbers). Nevertheless, I have reread the manuscript and (tried to) clarify where necessary. It is noted that this is an observational paper that hopefully triggers analytic and numerical modelling to better understand the relevant processes.

Thorpe scale analysis: There is a striking pattern in the calculated Thorpe displacements, indicating a very frequent and long-lasting 50m overturn at the lowest 50m. I assume this is an artefact, because the temperature gradient is often at or below 0.5mK/50m, and sensor noise and uncertainty are comparatively high. In such a constellation of a very low density gradient like in the BBL, noise/uncertainty will cause spurious overturns and overestimated displacements, leading to overestimated mixing through Thorpe scale analysis [Piera et al., 2002; Johnson and Garrett, 2004]. The diagnosed intense mixing in the bottom 50m is at the base of major results of the paper: the increasing turbulence with depth, intense near-bottom mixing, and the explanation for the intense near-bottom mixing by internal waves which trigger overturns in a marginal stable regime. Given the importance of the intense bottom boundary layer mixing for the paper, a critical review of the appropriateness of the used Thorpe scale processing should be a central part of the methods. If the existence of a quasipermanently overturning 50m-bottom-layer should prove true, this as well should be a central part of the discussions. »>I like to stress that the string of T-sensors is not a shipborne CTD, for many reasons. The mooring hardly moves, the 400 m profile is made within 0.02 s (instead of lowering a CTD-package at a speed of 0.8-1 m/s never making a correct vertical profile), such 400 m profile is made every 1 s providing many profiles to average over the buoyancy scale of one hour or more instead of a single CTD-cast, the observations are generally made in relatively high Reynolds number areas where the temperature density relationship is tight, and the NIOZ T-sensor have very low noise level about one-third of that of SeaBird 911 T-sensor. The two quoted papers focus on

noise by shipborne CTD, a completely different way of measuring than moored NIOZ-T (as Johnson and Garrett 2004 indeed indicate in their concluding remarks: noise problems may be very minor compared to other problems with shipborne CTD-data; Re: J&G2004 indicate underestimating of turbulence by noise, not overestimating). Ii is reminded (as was indicated in the text) that the resolvable dissipation rates by the moored T-sensors averaged over a 100-m vertical range is approximately 3ïĆť10-12 m2 s-3, much lower than resolvable by CTD (and generally much lower than dissipation rates observed in the lower 50 m of the range). This all is now made more explicit in Section 2, and with noise level panel added to Fig. B1. As for the turbulence in the lower 50 m of the range: no it is not quasi permanently, but slowly varying with time, dominantly on half the inertial period and on sub-inertial periodicities and sometimes on shorter timescales, as indeed indicated by this reviewer in the top-paragraph of this review. The text on this is all reread now and made more explicit where necessary.

Further remarks: - Given the reported numbers N = 5.5 * 10 ĚĘ -4, S = 1.6 * 10 ĚĘ -4 (lines 254 to 258), the average Richardson number in the BBL seems rather 10 than unity. This would not support the assumption of the BBL being systematically marginally stable. »>In the mean yes, but not in variability. It would have been better if current/shear observations were available over length scales of the thin layer stratification, but no such instrumentation was available. It was not clearly stated it was not to be systematically marginally stable 'to occur regularly' (now added 'in bursts').

- Can you make clearly understandable why the given arguments (lines 189 to 192) allow to choose a mixing efficiency parameter m of 0.2? »>It is following the works of Osborn, Dillon and Oakey (and many thereafter): after averaging over suitable number of profiles, length and time scales, this is the mean value to be found for mixing efficiency. Internal waves not only induce turbulent mixing by their breaking but also allow for rapid restratification making the mixing rather efficient.

- I'd propose to more prominently place the particular results for the abyssal hill region in the larger context of the limiting cases 'steep topography' and 'smooth abyssal plain'

»>OK, done now, with the restriction that in my view a smooth abyssal plain hardly exists: there are always topographic features; it's merely a matter of scale.

- data availability is not stated »>Done now

References: Piera, Roget, Catalan (2002): Turbulent patch identification in microstructure profiles: a method based on wavelet denoising and Thorpe displacement analysis, J. Atm. Oceanic Tech., 19, 1390-1402 Johnson and Garrett (2004): Effects of noise on Thorpe scales and run lengths, J. Phys. Oceanogr., 34, 2359-2372

Please also note the supplement to this comment:
https://www.biogeosciences-discuss.net/bg-2018-142/bg-2018-142-AC2-supplement.pdf

**Supplement:**

**Abyssal plain hills and internal wave turbulence**

**by Hans van Haren[1]**

[revised manuscript text omitted]
 mixing efficiency value is close to the tidal mean mixing potential observed by Cyr and van Haren (2016), also in layers in which stratification is weak.

Internal waves not only induce mixing through their breaking but also allow for rapid restratification, making the mixing rather efficient.

The moored T-sensor data are thus much more precise and apt for using Thorpe overturning scales to estimate turbulence parameters than shipborne CTD-data. Most of the concerns raised e.g. by Johnson and Garrett (2004) on this method using shipborne CTD-data are not relevant here. First, instead of a single (CTD-)profile, averaging is performed over $O(10^3-10^4)$ profiles, i.e. at least over the buoyancy time scale and more commonly over the inertial time scale. Second, the mooring is not moving more than 0.1 m vertically and if moving it does so on a sub-inertial time-scale:

No corrections are needed for 'ship motions' and instrumental/frame flow disturbance, as for CTD-data. Third, the noise level of the moored T-sensors is very low, about one- third of the high-precision sensors used in SeaBird 911 CTD (Appendix B). Fourth, the environment in which the observations are made is dominated by internal wave breaking (above topography), where turbulent mixing is generally not weak and where a tight temperature-density relationship exists. Because of points three and four, complex noise reduction as in Piera et al. (2002) is not needed for moored T-sensor data. More in general for these data in such environments, Thorpe overturning scales can be solidly determined using temperature sensor data instead of more imprecise density (T- and S-sensor data) as salinity intrusions are not found important as verified.

The buoyancy Reynolds number $Re_b = \varepsilon/\nu N^2$ is used to distinguish between areas of weak, $Re_b < 100$, and strong turbulence.

In the following, averaging over time is denoted by $[\ldots]$, averaging over depth- range by $<\ldots>$. The specific averaging periods and ranges are indicated with the mean values. The vertical coordinate z is taken upward from the bottom $z = 0$. Shear-induced overturns are visually identified as inclined S-shapes in log(N) panels while convection demonstrates more vertical columns (e.g., van Haren and Gostiaux, 2012; Fritts et al.,

2016). It is noted that both types occur simultaneously, as columns exhibit secondary shear along the edges and KHi demonstrate convection in their interior core (Li and Li,

2004; Matsumoto and Hoshino, 2006).

**3 Observations**

High-resolution T-sensor data analysis was difficult because of the very small temperature ranges and variations of only a few mK over, especially the lower, 100 m of the observed range. This rate of variation is less than the local adiabatic lapse rate.

First, a spectral analysis is performed to investigate the internal wave and turbulence ranges and slopes appearance. Then, particular turbulent overturning aspects of internal wave breaking are demonstrated in magnifications of time-depth series. Finally, profiles of mean turbulence parameter estimates are used to focus on the extent and nature of the bottom boundary layer.

.

**3.1 Spectral overview**

The small temperature ranges are reflected in the low values of the large-scale stratification (Fig. 2a). (Salinity contributes weakly to density variations, Appendix B).

Typical buoyancy periods are 3 h, increasing to roughly 9 h in near-homogeneous layers, e.g., near the bottom. In spite of the weak stratification, the IGW-band, approximately between and including f and N, is one order of magnitude wide. This

IGW-bandwidth is observable in spectra of turbulence dissipation rate (Fig. 2b) and temperature variance (Fig. 2c).

The T-sensors have identical instrumental (white) noise levels at frequencies $\sigma >$

$10^4$ cpd and near-equal variance at sub-inertial frequencies $\sigma < f$ (Fig. 2c). From the former an approximate one standard deviation is observed of std $\approx 4\times10^{-5}$ °C, see also

[revised manuscript text omitted]

Thus it seems that the precise characteristics (slopes/heights) of the hilly topography is not very relevant for the observed internal wave intensity and turbulence generation, as long as the bottom is not a flat plate and the hills have IGW-scales. This associates with the suggestion by Baines (2007) and Morozov (2018) that small-scale topography may prove not negligible for internal wave generation and dissipation in comparison with large oceanic ridges, seamounts and continental slopes. After all, flat bottoms do hardly exist in the ocean, depending on the length scale investigated.This probably holds for both the present observations in the stratified interior and those in the NBTZ.

The tenfold larger turbulence intensity observed here in the NBTZ compared to the stratified interiorin the latter marks a relatively extended inertial subrange. Although the near-bottom (6 mab) current speedmagnitudes are typically 0.05 m s$^{-1}$, up to about

0.10 m s$^{-1}$, the estimated turbulence intensity of $10^3$-$10^4$ times larger than molecular diffusion is sufficient to mix materials up to 100 mab, the extent of observed vertical mixing in the layer adjacent tot the bottom. This reflects previous observations of nephels, turbid waters of enhanced suspended materials (Armi and D'Asaro, 1980). It is expected that this material is resuspended locally, as the more intensely turbulent steeper large-scale slopes are too far away horizontally, far beyond the baroclinic

Rossby radius of deformation.

For the future, modelling may provide better insights in the precise coupling between near-inertial shear and internal wave breaking, leading to a combination of convective and shear-induced overturning. It is expected that interaction between the semidiurnal tidal current and the hilly topography may generate internal waves near the
buoyancy frequency (Hibiya et al., 2017), while it remains to be investigated whether
the inertial motions are shear are topographically or atmospherically driven. The one-
sided shear across thin-layer stratification, as inferred from observed deviation of high-
N sheets from isotherms and associated with the vertical propagation direction of
internal waves, may prove important for the wave breaking.

**5 Conclusions**

From the present high-resolution temperature sensor data moored up to 400 m above
a hilly abyssal plain in the northeastern Pacific we find an interaction between small-
scale internal wave propagation, large-scale near-inertial shear and the near-bottom
water phase. In an environment where semidiurnal tidal currents dominate, 200-m shear
is largest at the inertial frequency and near-bottom turbulence dissipation rates are
largest at twice the inertial frequency. Due to internal wave propagation and occasional
breaking, stratification in the overlying waters is organized in thin sheets, with less
stratified waters in larger layers in between, but turbulent erosion occurs
asymmetrically. The average amount of turbulent overturns due to internal wave
breaking here and there is equal to open ocean turbulence, with intensities about 100
times those of molecular diffusion. The high-frequency internal waves propagate to
near the bottom and likely trigger ten times larger turbulence there as shown in time-
average vertical profiles. The result is a highly variable near-bottom turbulent zone,
which may be near-homogeneous over heights of less than 7 m and up to 100 m above
the bottom. This near-bottom turbulence is not predominantly governed by frictional
flows on a rotating sphere as in Ekman dynamics that occupy a shorter range O(10 m)
above the bottom. Fronts occur and sudden isotherm-uplifts by solitary internal waves as well. Turbulence seems shear dominated, but occurs in parallel with convection. The shear is quasi-permanent because the dominant near-circular inertial motions have a constant magnitude. It is expected that inertial shear dominates also on shorter scales (not verifiable with the present current meter data), possibly added by smaller internal wave shear. In the mean, turbulence dissipation rate exceeds the level of $10^{-11}$ m$^2$s$^{-3}$, except for a 30 m thick layer around 100 mab. Given the numerous amount of hills distributed over the ocean floor, the present observations lend some support to their importance for internal wave turbulence generation in the ocean.

*Data availability.* Current meter and CTD data are stored in the JPIO-databank at Geomar Kiel,

Germany. The temperature sensor data are made available upon request to the author as they need to be computed from the raw data set for any given specific period.

[revised manuscript text omitted]

---

## Author Comment (AC3)

**Abyssal plain hills and internal wave turbulence**

**by Hans van Haren[1]**

[revised manuscript text omitted]

Internal waves not only induce mixing through their breaking but also allow for rapid restratification, making the mixing rather efficient.

The moored T-sensor data are thus much more precise and apt for using Thorpe overturning scales to estimate turbulence parameters than shipborne CTD-data. Most of the concerns raised e.g. by Johnson and Garrett (2004) on this method using shipborne CTD-data are not relevant here. First, instead of a single (CTD-)profile, averaging is performed over $O(10^3-10^4)$ profiles, i.e. at least over the buoyancy time scale and more commonly over the inertial time scale. Second, the mooring is not moving more than 0.1 m vertically and if moving it does so on a sub-inertial time-scale:

No corrections are needed for 'ship motions' and instrumental/frame flow disturbance, as for CTD-data. Third, the noise level of the moored T-sensors is very low, about one- third of the high-precision sensors used in SeaBird 911 CTD (Appendix B). Fourth, the environment in which the observations are made is dominated by internal wave breaking (above topography), where turbulent mixing is generally not weak and where a tight temperature-density relationship exists. Because of points three and four, complex noise reduction as in Piera et al. (2002) is not needed for moored T-sensor data. More in general for these data in such environments, Thorpe overturning scales can be solidly determined using temperature sensor data instead of more imprecise density (T- and S-sensor data) as salinity intrusions are not found important as verified.

The buoyancy Reynolds number $Re_b = \varepsilon/\nu N^2$ is used to distinguish between areas of weak, $Re_b < 100$, and strong turbulence.

In the following, averaging over time is denoted by […], averaging over depth- range by <…>. The specific averaging periods and ranges are indicated with the mean values. The vertical coordinate z is taken upward from the bottom z = 0. Shear-induced overturns are visually identified as inclined S-shapes in log(N) panels while convection demonstrates more vertical columns (e.g., van Haren and Gostiaux, 2012; Fritts et al.,

2016). It is noted that both types occur simultaneously, as columns exhibit secondary shear along the edges and KHi demonstrate convection in their interior core (Li and Li,

2004; Matsumoto and Hoshino, 2006).

**3 Observations**

High-resolution T-sensor data analysis was difficult because of the very small temperature ranges and variations of only a few mK over, especially the lower, 100 m of the observed range. This rate of variation is less than the local adiabatic lapse rate.

First, a spectral analysis is performed to investigate the internal wave and turbulence ranges and slopes appearance. Then, particular turbulent overturning aspects of internal wave breaking are demonstrated in magnifications of time-depth series. Finally, profiles of mean turbulence parameter estimates are used to focus on the extent and nature of the bottom boundary layer.

.

**3.1 Spectral overview**

The small temperature ranges are reflected in the low values of the large-scale stratification (Fig. 2a). (Salinity contributes weakly to density variations, Appendix B).

Typical buoyancy periods are 3 h, increasing to roughly 9 h in near-homogeneous layers, e.g., near the bottom. In spite of the weak stratification, the IGW-band, approximately between and including f and N, is one order of magnitude wide. This

IGW-bandwidth is observable in spectra of turbulence dissipation rate (Fig. 2b) and temperature variance (Fig. 2c).

The T-sensors have identical instrumental (white) noise levels at frequencies $\sigma >$

$10^4$ cpd and near-equal variance at sub-inertial frequencies $\sigma < f$ (Fig. 2c). From the former an approximate one standard deviation is observed of std $\approx 4 \times 10^{-5}$ °C, see also

[revised manuscript text omitted]

Zone (van Haren et al., 2014) and Kane Gap (van Haren et al., 2013) where current speeds are larger ~0.25 m s$^{-1}$. The through-flow data form a contrast with the present observations, because the near-bottom zone is stratified there despite relatively strong shear flow. They are similar in showing internal wave convection penetrating to close to the bottom, occasionally.

Sloping fronts are observed near the bottom in Armi and D'Asaro (1980)'s, Thorpe (1983)'s and the present data. However, isopycnal transport of mixed waters seems not away from the boundary as proposed in (Armi and D'Asaro, 1980) but rather into the

NBTZ sloping downward with time (present data). This governs the variable height of the NBTZ.

Although bottom slopes were about three times larger in the Northeast Pacific than above the Hatteras Plain, the present observations show many similarities as in Armi and D'Asaro (1980). They also show many similarities with equivalent turbulence estimates in both the interior and in the variable lower 100 mab compared with those from above the central Alboran Sea, a basin of the Mediterranean Sea (van Haren,

2015), and with observations made in the southeast Pacific abyssal hill plains around -

07° 07.213′ S, -088° 24.202′ W, East of the oriental Pacific Ridge (unpublished results).

Thus it seems that the precise characteristics (slopes/heights) of the hilly topography is not very relevant for the observed internal wave intensity and turbulence generation, as long as the bottom is not a flat plate and the hills have IGW-scales. This associates with the suggestion by Baines (2007) and Morozov (2018) that small-scale topography may prove not negligible for internal wave generation and dissipation in comparison with large oceanic ridges, seamounts and continental slopes. After all, flat bottoms do hardly exist in the ocean, depending on the length scale investigated.

The tenfold larger turbulence intensity observed here in the NBTZ compared to the stratified interior marks a relatively extended inertial subrange. Although the near- bottom (6 mab) current speeds are typically 0.05 m s$^{-1}$, up to about 0.10 m s$^{-1}$, the estimated turbulence intensity of $10^3$-$10^4$ times larger than molecular diffusion is sufficient to mix materials up to 100 mab, the extent of observed vertical mixing in the layer adjacent to the bottom. This reflects previous observations of nephels, turbid waters of enhanced suspended materials (Armi and D'Asaro, 1980). It is expected that this material is resuspended locally, as the more intensely turbulent steeper large-scale slopes are too far away horizontally, far beyond the baroclinic Rossby radius of deformation.

For the future, modelling may provide better insights in the precise coupling between near-inertial shear and internal wave breaking, leading to a combination of convective and shear-induced overturning. It is expected that interaction between the semidiurnal tidal current and the hilly topography may generate internal waves near the buoyancy frequency (Hibiya et al., 2017), while it remains to be investigated whether the inertial motions are shear are topographically or atmospherically driven. The one-sided shear across thin-layer stratification, as inferred from observed deviation of high-N sheets from isotherms and associated with the vertical propagation direction of internal waves, may prove important for the wave breaking.

**5 Conclusions**

From the present high-resolution temperature sensor data moored up to 400 m above a hilly abyssal plain in the northeastern Pacific we find an interaction between small-scale internal wave propagation, large-scale near-inertial shear and the near-bottom water phase. In an environment where semidiurnal tidal currents dominate, 200-m shear is largest at the inertial frequency and near-bottom turbulence dissipation rates are largest at twice the inertial frequency. Due to internal wave propagation and occasional breaking, stratification in the overlying waters is organized in thin sheets, with less stratified waters in larger layers in between, but turbulent erosion occurs asymmetrically. The average amount of turbulent overturns due to internal wave breaking here and there is equal to open ocean turbulence, with intensities about 100 times those of molecular diffusion. The high-frequency internal waves propagate to near the bottom and likely trigger ten times larger turbulence there as shown in time-average vertical profiles. The result is a highly variable near-bottom turbulent zone, which may be near-homogeneous over heights of less than 7 m and up to 100 m above the bottom. This near-bottom turbulence is not predominantly governed by frictional flows on a rotating sphere as in Ekman dynamics that occupy a shorter range O(10 m) above the bottom. Fronts occur and sudden isotherm-uplifts by solitary internal waves as well. Turbulence seems shear dominated, but occurs in parallel with convection. The shear is quasi-permanent because the dominant near-circular inertial motions have a constant magnitude. It is expected that inertial shear dominates also on shorter scales (not verifiable with the present current meter data), possibly added by smaller internal wave shear. In the mean, turbulence dissipation rate exceeds the level of $10^{-11}$ m$^2$s$^{-3}$, except for a 30 m thick layer around 100 mab. Given the numerous amount of hills distributed over the ocean floor, the present observations lend some support to their importance for internal wave turbulence generation in the ocean.

*Data availability*. Current meter and CTD data are stored in the JPIO-databank at

Geomar Kiel, Germany. The temperature sensor data are made available upon request to the author as they need to be computed from the raw data set for any given specific period.

[revised manuscript text omitted]

profiles gives $\langle\alpha\rangle = -0.223 \pm 0.005$ kg m$^{-3}$ °C$^{-1}$).

---

## Author Response (AR1)

**Nederlands Instituut voor Onderzoek der Zee**

NETHERLANDS INSTITUTE FOR SEA RESEARCH   (NIOZ)

Hans van Haren
P.O. BOX 59
1790 AB  DEN BURG
TEXEL - THE NETHERLANDS
TEL. (31) (0) 222 - 369300/369451
POSTBANKNO. 89505
e-mail  hans.van.haren@nioz.nl

**The Editor of Biogeosciences –**
**Special issue '**Assessing environmental impacts of
deep-sea mining – revisiting decade-old benthic
disturbances in Pacific nodule area**s'**

| ONZE REF. | : |
| UW REF | : |
| BETREFT | : |

Texel, 28/06/2018

Dear editor, Dear Dr. Haeckel,

I would like to submit for publication in BIOgeosciences, special issue on 'Assessing environmental impacts of deep-sea mining – revisiting decade-old benthic disturbances in Pacific nodule areas' the revised paper entitled,

" Abyssal plain hills and internal wave turbulence ".

All of your latest issues have been taken into account.

Yours sincerely,

Hans van Haren